

# A continuum model of ice mélange and its role during retreat of the Antarctic Ice Sheet

David Pollard[1], Robert M. DeConto[2], Richard B. Alley[1,3]

[1]Earth and Environmental Systems Institute, Pennsylvania State University, University Park, PA 16802, USA
[2]Department of Geosciences, University of Massachusetts, Amherst, MA 01003, USA
[3]Department of Geosciences, Pennsylvania State University, University Park, PA 16802, USA

*Correspondence to*: David Pollard (pollard@essc.psu.edu)

**Abstract.** Rapidly retreating thick ice fronts can generate large amounts of mélange (floating ice debris), which may affect episodes of rapid retreat of Antarctic marine ice. In modern Greenland fjords, mélange provides substantial back pressure on calving ice faces, which slows ice-front velocities and calving rates. On the much larger scales of West Antarctica, it is unknown if mélange could clog seaways and provide enough back pressure to act as a negative feedback slowing retreat. Here we describe a new mélange model, using a continuum mechanical formulation that is computationally feasible for long-term continental Antarctic applications. It is tested in an idealized rectangular channel, and calibrated very basically using observed modern conditions in Jakobshavn fjord, West Greenland. The model is then applied to drastic retreat of Antarctic ice in response to warm mid-Pliocene climate. With mélange parameter values that yield reasonable modern Jakobshavn results, Antarctic marine ice still retreats drastically in the Pliocene simulations, with little slowdown despite the huge amounts of mélange generated. This holds both for the rapid early collapse of West Antarctica, and later retreat into major East Antarctic basins. If parameter values are changed to make the mélange much more resistive to flow, far outside the range for reasonable Jakobshavn results, West Antarctica still collapses and retreat is slowed or prevented only in a few East Antarctic basins.

## 1 Introduction

Theory, modeling and observations point to the prospect of rapid grounding-line retreat and marine ice loss from West Antarctica and major East Antarctic basins, in response to climate warming (Weertman, 1974; Mercer, 1978; Schoof, 2007; Pritchard et al., 2012; Rignot et al., 2014). These rapid retreats are suspected to have contributed to high sea-level stands in the Pliocene and Pleistocene (Rovere et al., 2014; Dutton et al., 2015; Pollard et al., 2015; Sutter et al., 2016), and pose the threat of drastic sea-level rise due to future warming (Joughin et al., 2014a; Cornford et al., 2015, Feldmann and Levermann, 2015; Golledge et al., 2015; Ritz et al., 2015; DeConto and Pollard, 2016; Arthern and Williams, 2017). The retreats are thought to be amplified by runaway positive feedback mechanisms, termed Marine Ice Sheet Instability (MISI, Schoof, 2007) and/or Marine Ice Cliff Instability (MICI, Pollard et al., 2015; DeConto and Pollard, 2016), that occur in marine basins





whose bedrock topography deepens into the interior, due to the very strong dependence of ice export on grounding-line depth.

Calving of ice from thick (~1 km) glacial termini generates substantial amounts of floating ice debris called mélange, as observed in the fjords of major outlet glaciers of Greenland and in places in Antarctica (Macayeal et al., 1998; Joughin et al.,

2008, 2014b; Fricker et al., 2009; Khazendar et al., 2009; Amundson et al., 2010; Scambos et al., 2011). In Greenland, mélange occupies some or all of the fjords downstream of the ice terminus (Joughin et al., 2012; Sundal et al., 2013; Sutherland et al., 2014; Moon et al., 2015), and is thought to provide significant back pressure on the ice face, reducing calving and ice velocities especially in winter (Joughin et al., 2008, 2014b; Amundson et al., 2010; Walter et al., 2012; Todd et al., 2014). The lateral scales of these modern ice faces and fjords are 5 to 10 km. If large-scale retreat is initiated in the

Amundsen Sea sector of West Antarctica, for instance, the lateral scales of retreating grounding lines would potentially be an order of magnitude larger (100's km), flowing into relatively unconfined seaways, for which there is no modern analog. Vast amounts of mélange would presumably be generated, and it is unknown whether the mélange could act as a significant negative feedback, through clogging of seaways and back pressure on ice faces, reducing calving and slowing ice velocities and grounding-line retreat.

Here we formulate an explicit physically based model of mélange, and couple it to an existing ice sheet-shelf model. To date only a few studies have modeled mélange explicitly, and most use discrete-particle or granular-material approaches for ice and/or mélange (Bassis and Jacobs, 2013; Astrom et al., 2014; Peters et al., 2015; Robel, 2017). Discrete-particle models are potentially truer representations of real mélange (in Greenland fjords today, a poorly sorted agglomeration of ice pieces with sizes up to O(100 m), in a matrix of thin sea ice), but are computationally infeasible for the temporal and spatial scales

involved in Antarctic retreat.

Our approach is to use continuum physics, in a model that captures basic dependencies between rate of mélange supply, downstream export, side drag, and ocean bottom resistance, which combine to produce back pressure on ice faces. Two other continuum models have been applied to mélange to our knowledge (Seneca Lindsey and Dupont, 2012; Vankova and Holland, 2017), discussed briefly in section 3.

## 25 2 Relationship with Greenland fjords

We calibrate the model using basic observations of modern mélange in the Jakobshavn fjord. (For simplicity we use the term Jakobshavn throughout, and do not use separate names for the glacier and fjord, Jakobshavn Isbrae and Ilulissat Icefjord respectively). There are no comprehensive datasets of mélange properties in Jakobshavn or other fjords to our knowledge, but there are many individual studies with relevant observations and modeling (Joughin et al., 2008, 2014b; Amundson et al.,

2008; 2010, 2016; Motyka et al., 2011; Macayeal et al., 2012; Walter et al., 2012; Cook et al., 2013; Sundal et al., 2013; Foga et al., 2014; Foga, 2016; Sutherland et al., 2014; Todd et al., 2014; Cassotto et al., 2015; Krug et al., 2015; Moon et al., 2015; Murray et al., 2015; Enderlin et al., 2016). The main properties reported in these studies that are pertinent here, with quantities rounded to the nearest order of magnitude for Jakobshavn, are as follows.





- Mélange consists of discrete ice pieces, loosely cemented within sea ice, with the mass dominated by ice pieces, the largest of which are small compared to the fjord dimensions, as required for a continuum approach.

- Mélange is generated (supplied) by calving from a ~1 km thick grounded ice front. The ice velocity just upstream of the front is ~10 km yr$^{-1}$, and the mélange just downstream of the front is ~100 m thick. At the ice front, the km-scale calving pieces (ice bergs) must overturn and disintegrate very rapidly to maintain an initial mélange thickness of ~100 m. This occurs primarily in discrete events that episodically push the mélange downstream in rapid pulses. By conservation of mass, the mélange must move away from the ice front at an average speed ~10 times that of the incoming ice, thinning and/or slowing further down the fjord due to basal or surface melting.

- There is a pronounced seasonal cycle. In winter, the mélange is stiffened by the gluing effect of sea ice, enabling side drag to be transmitted as increased back pressure on the ice face, which prevents calving and allows a small floating ice tongue to form; the existing mélange and ice front move down the fjord together at the incoming ice speed (~10 km yr$^{-1}$). In summer, the mélange is more deformable, back stress is less, the ice tongue is lost, and episodic calving of ~km scale icebergs resumes, which pushes the mélange in the bulk of the fjord rapidly downstream in discrete pulses . On an annual mean the mélange moves down the fjord at tens of km yr$^{-1}$ to the mouth in Disko Bay, where it disperses and melts.

For now, we consider the ice and mélange state in only the modern Jakobshavn fjord, leaving past variations (Csatho et al., 2008; Joughin et al., 2008; Young et al.,2011) for possible future work. The model formulated below does not simulate discrete calving events, but rather the long-term average results of many such events. It does not simulate seasonal cycles of freezing/thawing sea ice and shutdown/resumption of calving, but represents the resulting annual average behavior. In particular there is no seasonal advance and retreat of model grounding lines, just annual mean motion.

At first sight this seems to pose a dilemma in applying the model to Jakobshavn and other Greenland fjords where the seasonal cycle plays a role, with mélange being pushed gradually downstream by the advancing glacier face in winter, and new mélange being created by episodic calving in summer that pushes existing mélange rapidly downstream in discrete pulses. The net effect is a horizontal "pump" that pushes the entire mélange body down the whole length of the fjord while the annual mean grounding-line position remains stationary. Despite the absence of a seasonal cycle, our model captures this "push-pump" mechanism via the boundary condition Eq. (B2) on mélange velocity at the ice interface, as described in Appendix B. As described below, mélange can also be driven downstream in the model by local hydrostatic pressure gradients, from thicker mélange at the head to thinner mélange at the mouth. Both the push-pump and pressure gradient mechanisms are active in our Jakobshavn simulations.

The model formulation is described in section 3, including discussion of the continuum mechanics approach. Idealized channel tests are performed in section 4.1, and the basic calibration vs. Jakobshavn is described in section 4.2. The model is applied to continental Antarctica in section 4.3, with simulations of drastic ice retreat during the warm mid Pliocene ~3 Ma. Conclusions on the role of mélange in Antarctic retreat are summarized in section 5.



### 3 Formulation

One possible choice for a continuum mechanical model of mélange is the viscous-plastic (VP) fluid formulation developed for sea ice (Hibler, 1977; Hibler and Tucker, 1979), which has been used and extended in many subsequent studies (e.g., Hunke et al., 1997) including a simplified cavitating fluid (CF) version (Flato and Hibler, 1992). Recently

Vasilova and Holland (2017) used cavitating-fluid dynamics in a continuum model of mélange. Their model is quite different from ours, explicitly incorporating large icebergs within a matrix of sea ice, and may be suited for smaller scales than considered here. For the continental and millennial scales of Antarctic ice retreat, the following considerations guided our choice of continuum model.

The VP-CF approach involves the concept of an internal pressure that resists convergence, which is an empirical function

of sea-ice thickness and represents ridging of ice slabs. As noted by Vasilova and Holland (2017), ridging is not relevant for mélange, but this function can still be used to represent resistance to convergence when large ice pieces in the mélange become closely packed. Instead of using a VP-CF approach, we use Shallow-Shelf Approximation (SSA) equations commonly used for ice shelf dynamics, modified to include (i) strong resistance to convergence beyond a certain "packing density", and (ii) very little resistance to divergence. With these modifications, the resulting system functions quite similarly

to a CF model, but we feel it has some advantages including adjustable non-linear rheology, and inclusion of hydrostatic pressure gradients, as described below.

In this preliminary study, for simplicity we use only one variable to represent mélange amount, $h_m$. It is called "thickness" below, but can be interpreted as a combination of thickness and density of the ice pieces responsible for most of the stresses in the mélange. A second prognostic variable could be added to represent compactness or fractional cover as in

sea-ice models ($A$ in Hibler, 1977; Flato and Hibler, 1992), but given the uncertainties in mélange rheology, we choose to start with just one variable $h_m$. Strong resistance to convergence beyond a certain packing density is included as a pressure term $P_p$ in the equations below, that is zero for small $h_m$ and ramps up strongly as $h_m$ exceeds a certain value.

If mélange never overrode itself so as to increase its bulk thickness, $P_p$ would be the only pressure term needed. This appears to be the case for the largest bergs embedded in Greenland mélange (~100's m scale, Enderlin et al., 2016), for

which compressive forces are never strong enough to cause overriding. However, overriding is conceivable for smaller pieces, and may be more common on the much larger scales of Antarctica. This process is represented in the equations below simply by using the vertically integrated hydrostatic pressure gradient within the mélange.

New mélange is supplied by calving or cliff failure of a solid ice face, added to the adjacent grid cell. This is either pushed downstream by the moving ice face (see Appendix B), or piles up locally, increasing the local pressure and the

velocities away from the face, so that a balance between supply and downstream advection is reached.

Following the considerations discussed above, we utilize and adapt the Shallow-Shelf Approximation (SSA) scaled equations, used in many studies to describe ice shelf and ice stream flow with very little basal drag, in which nearly all of the flow is due to horizontal stretching, and vertical shear is negligible. Seneca Lindsey and Dupont (2012) also used SSA





equations in a model of mélange, with much the same motivation as here; however, their study was limited to idealized channels, and they used much smaller contrast between mélange and ice rheology than here, and other simplifications (see journal discussion).

The starting point for SSA scaling from more primitive equations is the constitutive relation between deviatoric stresses

$\tau'$ and strain rates for polycrystalline ice (e.g., Thoma et al., 2014), modified for mélange here by the factor $f$ in the diagonal terms that reduces resistance to divergence (see below).

$$\tau' = 2\eta \begin{pmatrix} f\dfrac{\partial u}{\partial x} & \dfrac{1}{2}\left(\dfrac{\partial u}{\partial y}+\dfrac{\partial v}{\partial x}\right) & \dfrac{1}{2}\left(\dfrac{\partial u}{\partial z}+\dfrac{\partial w}{\partial x}\right) \\[3mm] \dfrac{1}{2}\left(\dfrac{\partial u}{\partial y}+\dfrac{\partial v}{\partial x}\right) & f\dfrac{\partial v}{\partial y} & \dfrac{1}{2}\left(\dfrac{\partial v}{\partial z}+\dfrac{\partial w}{\partial y}\right) \\[3mm] \dfrac{1}{2}\left(\dfrac{\partial u}{\partial z}+\dfrac{\partial w}{\partial x}\right) & \dfrac{1}{2}\left(\dfrac{\partial v}{\partial z}+\dfrac{\partial w}{\partial y}\right) & f\dfrac{\partial w}{\partial z} \end{pmatrix} \tag{1}$$

where $\eta$ is the effective viscosity

$$\eta = \tfrac{1}{2}(EA)^{-1/n}\,\dot{\varepsilon}^{(1-n)/n} \tag{2}$$

$A$ in (2) is the Arrhenius rate coefficient, $E$ is a dimensionless flow-enhancement factor, and $n$ is the rheological exponent, all specified below. $\dot{\varepsilon}$ is the effective strain rate (second tensor invariant) given by the individual strain rates $\dot{\varepsilon}_{ij}$

$$\dot{\varepsilon}^2 = \dot{\varepsilon}_{xx}{}^2 + \dot{\varepsilon}_{yy}{}^2 + \dot{\varepsilon}_{xx}\dot{\varepsilon}_{yy} + \dot{\varepsilon}_{xy}{}^2 + \dot{\varepsilon}_{xz}{}^2 + \dot{\varepsilon}_{yz}{}^2$$

$$\approx \left(\frac{\partial\overline{u}}{\partial x}\right)^2 + \left(\frac{\partial\overline{v}}{\partial y}\right)^2 + \frac{\partial\overline{u}}{\partial x}\frac{\partial\overline{v}}{\partial y} + \frac{1}{4}\left(\frac{\partial\overline{u}}{\partial x}+\frac{\partial\overline{v}}{\partial y}\right)^2 + \frac{1}{4}\left(\frac{\overline{\partial u_i}}{\partial z}\right)^2 + \frac{1}{4}\left(\frac{\overline{\partial v_i}}{\partial z}\right)^2 \tag{3}$$

The last two terms in (3) roughly represent vertical shear from the hybrid combination with SIA flow in Pollard and

DeConto (2012), and do not enter for mélange.

The $f$ terms in (1) propagate straightforwardly through the steps for SSA scaling (e.g., Thoma et al., 2014), yielding the equations for SSA velocities $u_m$ and $v_m$:

$$\frac{\partial}{\partial x}\left[f\,2\eta h_m\left(2\frac{\partial u_m}{\partial x}+\frac{\partial v_m}{\partial y}\right)\right] + \frac{\partial}{\partial y}\left[\eta h_m\left(\frac{\partial u_m}{\partial y}+\frac{\partial v_m}{\partial x}\right)\right] = \frac{\partial P_p}{\partial x} + \rho_m g h_m\frac{\partial h_s}{\partial x} + \beta u_m \tag{4a}$$

$$\frac{\partial}{\partial y}\left[f\,2\eta h_m\left(2\frac{\partial v_m}{\partial y}+\frac{\partial u_m}{\partial x}\right)\right] + \frac{\partial}{\partial x}\left[\eta h_m\left(\frac{\partial u_m}{\partial y}+\frac{\partial v_m}{\partial x}\right)\right] = \frac{\partial P_p}{\partial y} + \rho_m g h_m\frac{\partial h_s}{\partial y} + \beta v_m \tag{4b}$$

where $h_m$ is mélange thickness, and $h_s$ is its surface elevation. $\rho_m$ and $\rho_w$ are densities of mélange and sea water respectively, and g is the gravitational acceleration. $\rho_m g\, h_m\,[\partial h_s/\partial x,\,\partial h_s/\partial y]$ is the vertically integrated hydrostatic pressure gradient in the





mélange column (called the "driving stress" in ice sheet and shelf dynamics). Averaged over sufficiently wide area, columns of individual stacked pieces must be at or very close to flotation as a whole, i.e., the mélange extends from $(1- \rho_m/\rho_w)\, h_m$ above the ocean surface to $(\rho_m/\rho_w)\, h_m$ below, so that the driving stress is equal to $(1- \rho_m/\rho_w)\, \rho_m\, g\, h_m\, [\partial h_m/\partial x,\ \partial h_m/\partial y]$. Sea-water density $\rho_w = 1024$ kg m$^{-3}$, solid ice density (used below) $\rho_i = 910$ kg m$^{-3}$, and the bulk mélange density $\rho_m = 930$ kg m$^{-3}$

allowing for some liquid between the solid ice pieces.

$P_p$ in Eq. 4 is the vertically integrated pressure term resisting convergence beyond a certain packing density (represented loosely by the single variable $h_m$ as discussed above), given by

$$P_p = \rho_m \left(1 - \rho_m/\rho_w\right) gH_p^{\,2} \max\left[h_m - H_p, 0\right] / 10 \tag{5}$$

where $H_p$ is a constant representing the value of $h_m$ above which packing of the largest ice pieces in the mélange becomes

significant. $P_p$ is zero for $h_m < H_p$, and increases rapidly for every 10-meter increment above $H_p$, scaled by the effective vertically integrated hydrostatic pressure. In the experiments below, $H_p$ ranges from 30 m to 200 m, and is always somewhat greater than the thickness of newly created mélange ($H_n$ in Eq. B3; see Appendix B).

$\beta u_m$ and $\beta v_m$ in Eq. 4 are basal drag components. If the mélange grounds on the ocean bed, sliding occurs, with the linear coefficient $\beta = 0.01$ Pa m$^{-1}$ yr. If the ocean is deeper than the mélange base (the usual case), a small amount of water friction

is applied, linearly dependent on the ice velocity, with $\beta = 10^{-7}$ Pa m$^{-1}$ yr. This value is guided by earlier studies of sea ice dynamics (e.g., Hibler and Tucker, 1979), but increased by several times to allow for rougher mélange base and form drag (Hunke et al., 1997). Ocean currents are neglected, as are wind stress, sea-surface slopes associated with ocean circulation, and Coriolis terms, but all could be added straightforwardly in further work as in sea-ice models.

The boundary condition for (4) at an open ocean relates strain rates $\partial u_m/\partial x$ or $\partial v_m/\partial y$ to the mélange face thickness, just

as for SSA (MacAyeal, 1996; Thoma et al., 2014) with the additional $f$ term. At an ocean boundary perpendicular to the $x$ direction for instance,

$$f\, 2\eta h_m \left( 2\frac{\partial u_m}{\partial x} + \frac{\partial v_m}{\partial y} \right) = \frac{\rho_m g h_m^{\,2}}{2} \tag{6}$$

and similarly for a boundary perpendicular to the $y$-direction boundary except with $2\partial v_m/\partial y + \partial u_m/\partial x$ on the left side. In the model, the mélange usually thins outward to small thicknesses (meters) with further extension prevented by atmospheric or

oceanic melting. At an ice face, there is a boundary condition for mélange velocity perpendicular to the face (Eq. B2, derived in Appendix B), that captures the "push-pump" action of the face as mentioned above. For flow parallel to adjacent land or ice, a parameter $S$ is used to set side friction, ranging from no slip ($S = 1$) to free slip ($S = 0$), i.e., the drag per unit length for flow along a boundary parallel to the $x$-axis for instance is $S\, h_m\, \eta\ \partial u_m/\partial y$.

The choice of rheological exponent $n$ in the effective viscosity is very uncertain. The micro-physical processes that make

$n=3$ appropriate for polycrystalline ice are not relevant for mélange, but if analogies with deformation of other granular materials such as till are relevant, larger values may be more realistic (Rathbun et al., 2008). The various runs in this paper use $n$ values of 1, 5, and 10. The Arrhenius coefficient $A$ in Eq. 2 for temperate ice is $0.6 \times 10^{-8}$ Pa$^{-1}$ yr$^{-1}$ for $n=1$, $0.6 \times 10^{-24}$



Pa$^{-5}$ yr$^{-1}$ for $n$=5, and 0.6 x 10$^{-44}$ Pa$^{-10}$ yr$^{-1}$ for $n$=10. These values (multiplied by the enhancement factor $E$) are used in the idealized channel tests below. For Jakobshavn and Antarctica, they are modified depending on ice temperature deduced from surface air climatology (similarly to Pollard and DeConto, 2012).

As mentioned above, the $f$ term in Eqs. 4 and 6 is used to strongly decrease resistance to divergence, as appropriate for a granular material. It is 1 if $\partial u_m/\partial x < 0$, or 0.1 if $\partial u_m/\partial x > 0$ (where it multiplies $\partial u_m /\partial x$, and similarly for $\partial v_m /\partial y$). The value of 0.1 is admittedly arbitrary, and could be chosen smaller, or zero. However, much smaller values than ~0.1 caused numerical instabilities in large-scale simulations. The basic effect of changing $f$ is similar to changing the flow enhancement factor $E$, at least for divergent flow, and the latter is included in the sets of runs below.

The main prognostic equation for mélange thickness $h_m$, expressing conservation of mass, is

$$\frac{\partial h_m}{\partial t} + \frac{\partial(u_m h_m)}{\partial x} + \frac{\partial(v_m h_m)}{\partial y} = M - O + B$$

10                                                                                                 (7)

where $O$ is oceanic basal melt, and $B$ is atmospheric net surface budget (mainly snowfall minus melt), computed or prescribed as for ice shelves in the ice model (Pollard and DeConto, 2012; Pollard et al., 2015). $M$ is a supply term representing generation of mélange by calving or structural failure of ice faces, and is applied only to mélange (oceanic) grid boxes immediately adjacent to these ice faces.

Equations (1) to (7) are essentially the same as for ice shelves except for the added $f$ terms and $P_p$, and are solved numerically as in Pollard and DeConto (2012). The same Arakawa-C grid is used as in the ice model, with $u_m$ and $v_m$ staggered half a grid box in the $x$ and $y$ directions respectively. As in Pollard and DeConto (2012), upstream finite differencing is used for the advective terms $u_m h_m$ and $v_m h_m$ in (6). Up to three inner Picard iterations are performed between (2,3) and (4) to allow for the dependence of $\eta$ and $f$ on velocities, and up to five outer Runge-Kutta iterations are performed

at each time step between (4) and (7).

The mélange model is coupled as an additional component into our current ice sheet-shelf model (DeConto and Pollard, 2016). It runs on the same horizontal grid (longitude-latitude for Greenland, polar stereographic for Antarctica). Rates of calving and cliff failure at ice faces, computed in the ice model, are passed as input to the mélange model, and the mélange model passes the back stress of mélange on these faces back to the ice model. The calculation of back stress of mélange on

ice faces involves the rate of divergence in the mélange adjacent to the face relative to its free floating value, just as for ice shelves at grounding lines (Schoof, 2007; Pollard and DeConto, 2012). The back-stress calculation is a bit more involved, however, because the mélange occupies only a portion of the vertical ice face; the expression used is derived in Appendix A.

The faster speeds of mélange require much shorter timesteps than for ice sheet and shelf dynamics, and long-term Antarctic simulations are practically feasible only at coarse (40 km) resolution. However, idealized tests shown below

suggest that results depend only slightly on model resolution.





## 4 Results

### 4.1 Rectangular channel

As a preliminary 2-D test, the mélange model is applied to an idealized rectangular channel, 300 km long and 100 or 50 km wide. A Cartesian grid is used with 10x10 km resolution; as shown below, the results are very similar at resolutions

5   ranging from 20 to 2 km. The model is not coupled to the ice sheet model; instead, a supply of ice is prescribed flowing from the left, with ice velocity 5000 m yr$^{-1}$ and thickness 500 m, calving into the left hand edge of the domain. Oceanic melt at the base of the mélange is set to 15 m yr$^{-1}$, with zero surface mass balance of snowfall and snowmelt.

The model is initialized with no mélange, and run for 300 years to equilibrium. Results are shown in Fig. 1 for various combinations of mélange parameters. As expected, the mélange is thickest adjacent to or near the calving front, and thins

10   downstream. The combined pushing by the ice face and the mélange surface slope drives downstream velocities, exporting the supply at the calving front. Oceanic melt increases the rate of downstream thinning, and the mélange thins to nearly zero at which point it cannot advance one more grid cell given the ocean melt rate (there is no sub-grid fractional area for mélange, discussed further in section 5).

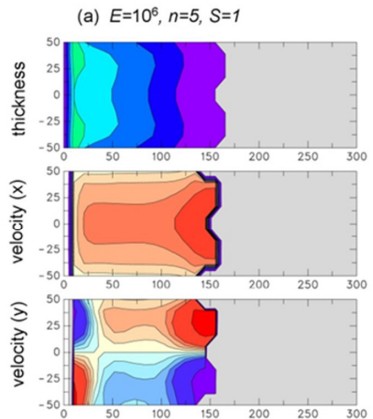
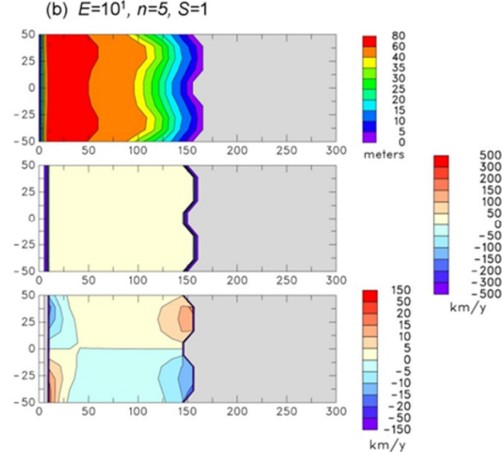





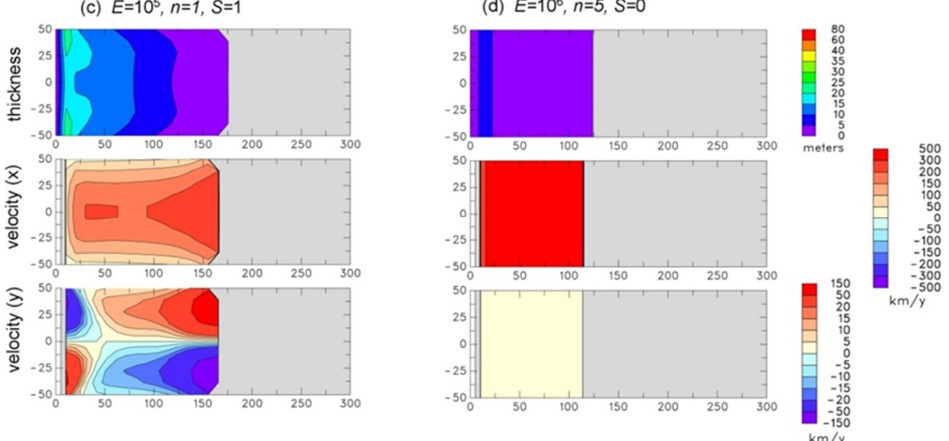

**Figure 1.** Maps of mélange thickness $h_m$ (upper, m), downstream velocity $u_m$ (middle, km yr⁻¹) and transverse velocity $v_m$ (lower, km yr⁻¹) in idealized channel simulations with different mélange parameter settings. Mélange flow is left to right, driven by prescribed supply at the left-hand edge. Grey regions are open ocean. The channel length is 300 km and width is 100 km. **(a)** with flow enhancement factor $E=10^6$, rheological exponent=5, side drag coefficient $S=1$. **(b)** as (a) except $E=10^1$. **(c)** as (a) except $E=10^5$ and $n=1$. **(d)** as (a) except $S=0$. In all cases, new mélange thickness $H_n$=30 m and pressure-scaling thickness $H_p$=60 m (in Eqs. B3 and 5 respectively).

Fig. 1a shows results for the nominal set of parameters used throughout the paper (producing the near-best score in the Jakobshavn ensemble further below):

- $E = 10^6$, flow enhancement factor in setting of effective viscosity in Eq. 2.
- $H_n = 30$ m, thickness of newly created mélange in Eq. B3.
- $H_p = 60$ m, pressure-scaling thickness for packing pressure in Eq. 5.
- $n = 5$, rheological exponent in Eq. 2.

The resulting mélange is ~20 m thick at the ice face, thinning uniformly and accelerating downstream, with fastest velocities of ~200 km yr⁻¹ at the downstream edge. There is a secondary circulation in the transverse direction, much slower than the downstream flow. It produces divergence away from most of the centerline, which may explain why the downstream mélange edge is bowed slightly upstream near the center.

In Fig. 1b, the flow enhancement factor $E$ is reduced to $10^1$. As expected, the mélange is much thicker, ~70 m at the ice face, and velocities are reduced to ~40 km yr⁻¹. In Fig. 1c, the rheological exponent $n$ is reduced to 1 corresponding to linear viscosity (and $E$ is adjusted slightly), which yields mélange thicknesses and velocities quite similar to Fig. 1a ($n$=5), although there is no bowing of the downstream mélange edge. Fig. 1d shows the effect of free-slip lateral boundaries ($S$=0), for which



the mélange is only ~6 m at the ice face, and downstream velocities are ~450 km yr$^{-1}$. Some of these basic sensitivities will be seen in the Jakobshavn and Antarctic simulations below.

Table 1 shows other quantities for the runs in Fig. 1. As expected, the net additional back force on the ice face ($\Delta F$) increases (decreases) for stiffer (weaker) mélange and more (less) side-drag influence.

| panels in Fig. 1 | $E$ | $n$ | $S$ | $\Delta F$ (Newtons) | $\theta_m(left)$ | $h_m(left)$ (m) |
|---|---|---|---|---|---|---|
| (a) | $10^6$ | 5 | 1 | $0.184 \times 10^{11}$ | 0.142 | 22.1 |
| (b) | $10^1$ | 5 | 1 | $0.658 \times 10^{12}$ | 0.109 | 75.5 |
| (c) | $10^5$ | 1 | 1 | $0.115 \times 10^{11}$ | 0.431 | 20.8 |
| (d) | $10^6$ | 5 | 0 | $0.536 \times 10^9$ | 0.677 | 6.18 |
| $\beta$=0 (not shown) | $10^6$ | 5 | 0 | 0.0 | 1.0 | 4.46 |

**Table 1.** Quantities for channel simulations in Fig. 1. Prescribed quantities are: $E$ = flow enhancement factor, $n$ = rheological coefficient, and $S$ = side drag coefficient. Resulting quantities are: $\Delta F$ = additional back force on the left-hand ice face due to mélange, compared to that due to ocean water pressure with no mélange, integrated over the width of the channel (see
10    Appendix A); $\theta_m(left)$ = factor representing degree of buttressing in the mélange, averaged over the left-hand ice face ($\theta_m = 1$ for free flow, $\leq 0$ if fully buttressed; see Eqs. A13, A15); $h_m(left)$ = mélange thickness averaged over the left-hand ice face. The last row shows an additional run with no ocean water drag ($\beta$=0 in Eq. 4), so there are no retarding external forces on the mélange at all, for which the resulting $\Delta F$=0 and $\theta_m$=1 as expected.

15    Fig. 2 shows that results depend reasonably little on grid size, at least for an idealized channel. This feature is important given the relatively coarse resolutions used in the Antarctic simulations below.





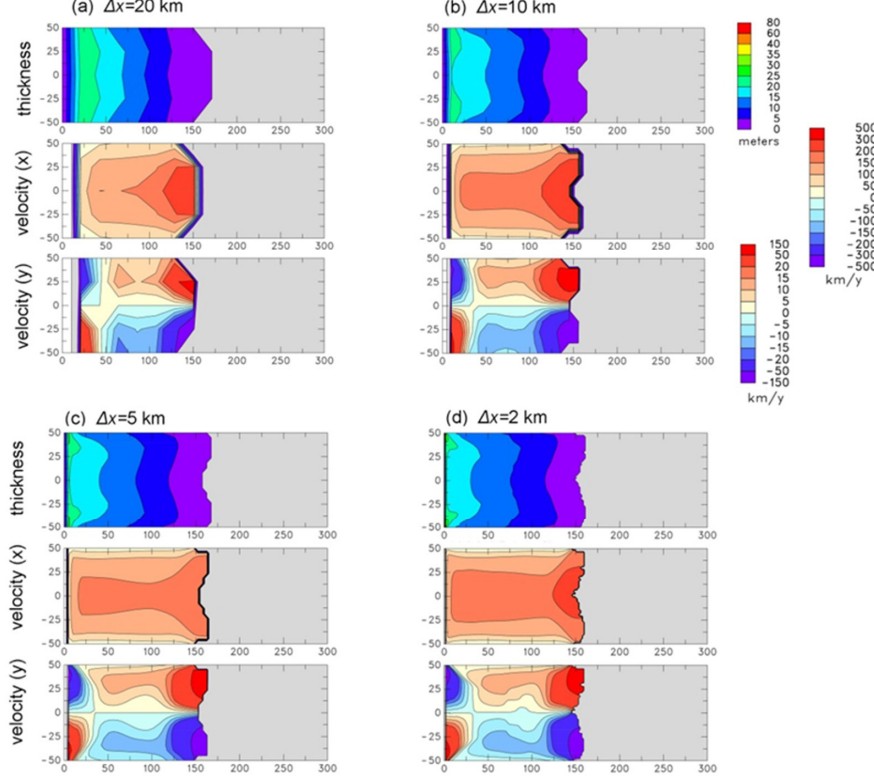

**Figure 2**. Same as Fig. 1a (*E*=10⁶, *S*=1), for different model grid sizes. **(a)** 20 km. **(b)** 10 km, same as Fig. 1a. **(c)** 5 km. **(d)** 2 km.

5    **4.2 Jakobshavn fjord**

The modern state of mélange and ice in the Jakobshavn fjord of West Greenland is used as a basic calibration of the model. This exercise is not intended as a full-blown modeling study of Jakobshavn, not least because the model resolution of 2 km barely resolves lateral fjord features. The intent here is just to establish very rough constraints on important mélange parameters, with a resolution barely resolving the geometry of interest as in the simulations of Antarctic basins in the next

10   section.

The coupled ice-sheet-shelf-mélange model is run in nested mode (Pollard and DeConto, 2012), over a longitude-latitude region bounded by 51.83 to 47.83 ºW, 68.42 to 69.92 ºN. This is roughly a 160 x 160 km rectangle centered on the Jakobshavn fjord, and extending far enough to the north, south and east to provide a sufficient buffer from the domain





boundaries. The figures below show a zoomed-in subset of the domain over the fjord itself. Lateral ice boundary conditions (ice thickness, velocities and temperatures) at the domain boundaries are provided by a previous simulation of modern continental Greenland (with no mélange component) at 0.1 ° latitude x 0.2 ° longitude resolution. The nested runs have a resolution of 0.02 ° latitude and 0.04 ° longitude (roughly 2x2 km). Numerical stability is an issue for these higher

resolutions. Some regions of parameter space with weak mélange become dynamically unstable, and this becomes more pervasive at 1 km resolution. With 2 km resolution, only a few runs are unstable (as shown below), and there are enough stable simulations to broadly map parameter space and constrain the basic ranges.

The nested model is initialized to the modern ice state interpolated from the continental run and no mélange, and run 50 years to equilibrium. Both for the continental and nested runs, climatological monthly surface air temperatures and

precipitation used for surface mass balance are from the RACMO2 regional model (van Angelen et al., 2014), and ocean temperatures for the sub-ice melt parameterization are from Levitus et al. (2012), although the 400-m depth used for water temperatures as in Pollard et al. (2015) may not be as appropriate for Greenland. The modeled ice margins in the upstream Jakobshavn region are reasonably realistic (Bamber et al., 2013), as is the location of the grounding line at the head of the fjord, and mélange is generated both by cliff failure on the northern side and calving of a very small ice shelf on the southern

side. The modeled surface mass balance on the Jakobshavn mélange is around -5 m yr$^{-1}$, mainly due to summer melt, which has an insignificant effect on total downstream mélange flux due to the short residence time in the fjord.

However, further ad-hoc adjustments to the ocean sub-ice melting are needed to achieve rough realism in the nested simulations. To keep the grounding line from advancing too far, oceanic basal melting of the ice shelf at the head of the fjord is prescribed to be 200 m yr$^{-1}$. For mélange, basal ocean melting in the fjord is set to 30 m yr$^{-1}$, which is near the low end but

much less than the high end (~300 m yr$^{-1}$) of ranges estimated by Enderlin et al. (2016). As discussed there, fjord waters are stratified by salinity, with near-freezing water at the top, and warmer and more saline waters below penetrating from Disko Bay (Holland et al., 2008; Motyka et al., 2011; Gladish et al., 2015). This vertical temperature gradient should cause more basal melting for larger ice pieces, but here we simply impose a uniform value. As the mélange approaches Disko Bay, basal melting ramps up to 200 m yr$^{-1}$ (using the arc-to-open-ocean parameterization that modifies Levitus-calculated melt rates;

Pollard and DeConto, 2012; Pollard et al., 2015).

The first two experiments in Fig. 3a and 3b (first two columns of panels) show that rough agreement with the modern Jakobshavn state can be achieved with appropriately chosen mélange parameters. These combinations of parameters produce near-best scoring in the Jakobshavn ensemble shown further below. The overall magnitudes of mélange thicknesses, downstream velocities, and east-west extent correspond with observed fjord-wide average values: 50 to 150 m for thickness

(Enderlin et al., 2016), 20 to 60 km yr$^{-1}$ for velocities (Sundal et al., 2013; Foga et al., 2014; Amundson et al., 2016; Enderlin et al., 2016), and 40 to 70 km for total length. These observational ranges are discussed further in Appendix C. In most simulations there are small areas where the mélange grounds on bedrock, along the sides of the fjord and at its head, which slightly slow the mélange and increase back stress at the ice face. Simulated downstream centerline velocities accelerate to several tens of km yr$^{-1}$ at mid-fjord, with faster velocities at the mouth. The net additional back force due to mélange




(compared to open water) on solid ice faces at the head of the fjord is 0.25 x $10^{12}$ N, or 31 kPa averaged over 8.1 x $10^6$ $m^3$ of ice face, which is comparable to that estimated for the Store Glacier in West Greenland (30 to 60 kPa; Walter et al. 2012; Todd et al., 2015). Note that the total downstream flux of mélange (~0.25 x $10^{11}$ $m^3$ $yr^{-1}$) at the head of the fjord is ~50% smaller than the observed Jakobshavn basin discharge (Howat et al., 2011, Supp. Inf.), mainly as a consequence of under-

5    resolved grounding-zone bathymetry.

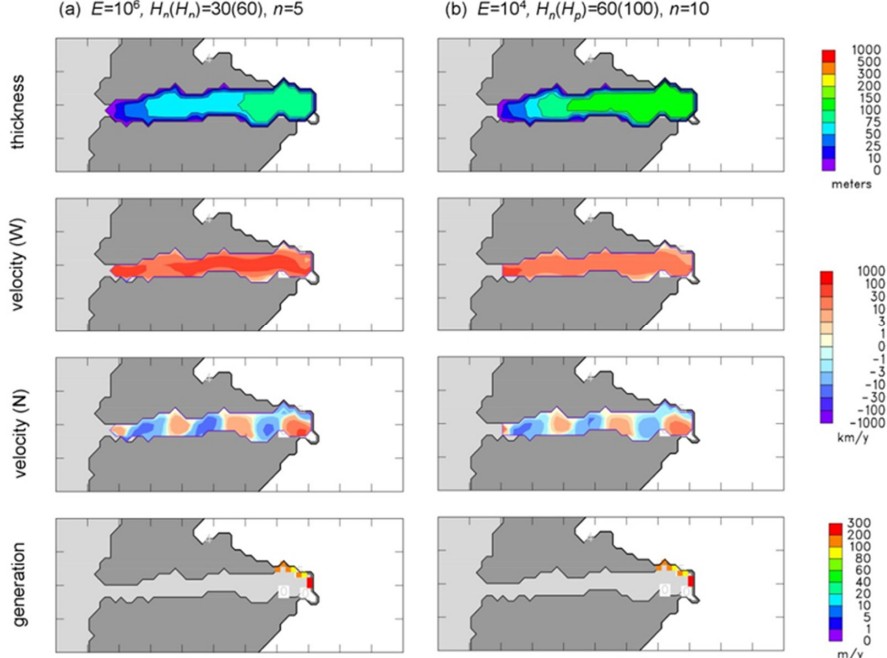



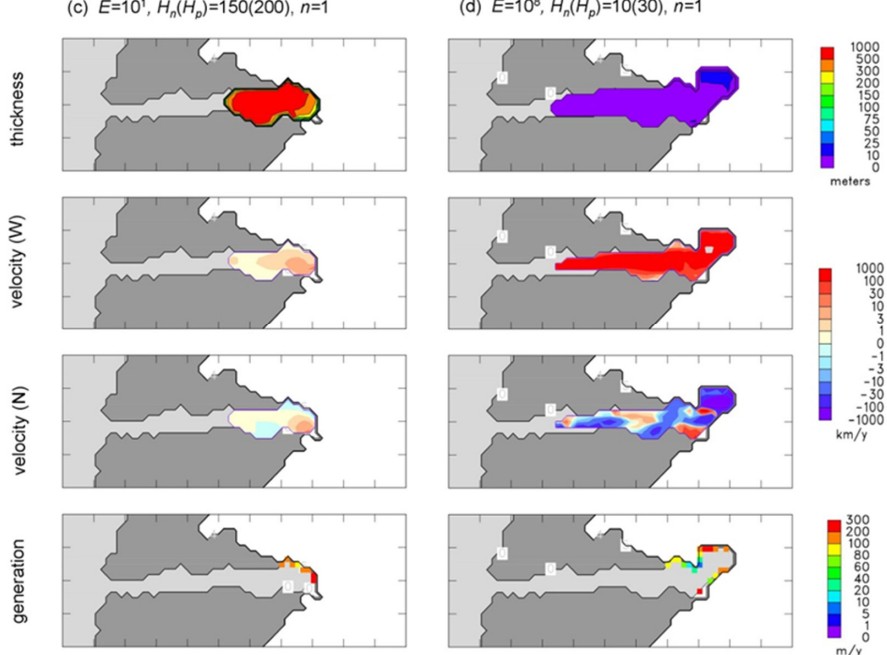

**Figure 3.** Maps of mélange quantities in nested simulations of the Jakobshavn region of West Greenland with the coupled ice-mélange model, for three different mélange parameter settings. Flow is right to left. White regions are grounded ice sheet, with a small ice shelf tongue in the southeastern corner of the fjord head. Dark grey regions are ice-free land or inland lakes, and light grey is ocean (Disko Bay, and mélange-free fjord water except in the bottom row). Axis tick marks are roughly 10 km apart. **Column (a):** with $E=10^6$, $H_n(H_p)=30(60)$, $n=5$ (near-best scoring run in the ensemble below). **Column (b):** with $E=10^4$, $H_n(H_p)=60(100)$, $n=10$ (another relatively realistic run). **Column (c):** with $E=10^1$, $H_n(H_p)=150(200)$, $n=1$(very stiff and thick mélange). **Column (d):** with $E=10^8$, $H_n(H_p)=10(30)$, $n=1$(very weak and thin mélange). **Top row:** mélange thickness (m). **Second row:** westward velocity (km yr$^{-1}$). **Third row:** northward velocity (km yr$^{-1}$). **Bottom row:** rate of mélange generation due to ice calving and/or cliff failure, supplied only to mélange grid boxes adjacent to ice faces (m yr$^{-1}$ of mélange thickness, $M$ in Eq. 7).

Fig. 3c (third column) shows an unrealistic simulation, in which the mélange viscosity, and new and pressure-scaling mélange thickness values have been increased, and $n$ set to 1, which produces much stiffer and thicker mélange. Mélange thicknesses are ~1000 m, and the mélange becomes grounded nearly everywhere on the fjord bed, extending only ~25 km down the fjord. Partly as a consequence of the increased basal drag, centerline velocities are only ~1 to 3 km yr$^{-1}$.





The opposite unrealistic situation is shown in Fig. 3d (last column), with greatly reduced viscosity, very thin new and scaling mélange thickness values, and $n=1$. The mélange is only a few meters thick, and downstream velocities are several hundred km yr$^{-1}$. Because of the greatly reduced back pressure on the ice face, the grounding line has retreated about 10 km into the interior at the northeastern end of the fjord.

5    To efficiently map out the simulated Jakobshavn behavior over parameter space, we performed an ensemble of simulations for all combinations of ranges of three selected mélange parameters: viscosity (via enhancement factor $E$ in Eq. 2), thickness of newly created mélange $H_n$ in Eq. B3 (matched with $H_p$ in Eq. 5), and rheological exponent $n$ in Eq. 2. Each simulation is run for 50 years to equilibrium as above, 75 runs in all. For each simulation, a score is computed that very roughly represents the realism of the result, combining average departures from observed magnitudes of mélange thicknesses, velocities and extent in the modern fjord. Details of the scoring calculation are described in Appendix C.

The score results for the whole ensemble are shown in Fig. 4. Realistic results (low scores) are achieved within fairly narrow ranges of enhancement coefficient ($E \sim 10^4$ to $10^6$) and thickness scales ($H_n \sim$30 to 60, $H_p \sim$ 60 to 100), but for a wide range of rheological exponent ($n \sim 1$ to 10). Outside these ranges of $E$ and $H_n(H_p)$, the modeled mélange is generally much too thick and slow as in Fig. 3c, or much too thin and fast. The along-fjord profiles of modeled mélange thickness and velocity in Fig. 5 illustrate the wide range of results, and how reasonable magnitudes are only obtained for limited ranges of parameter values.

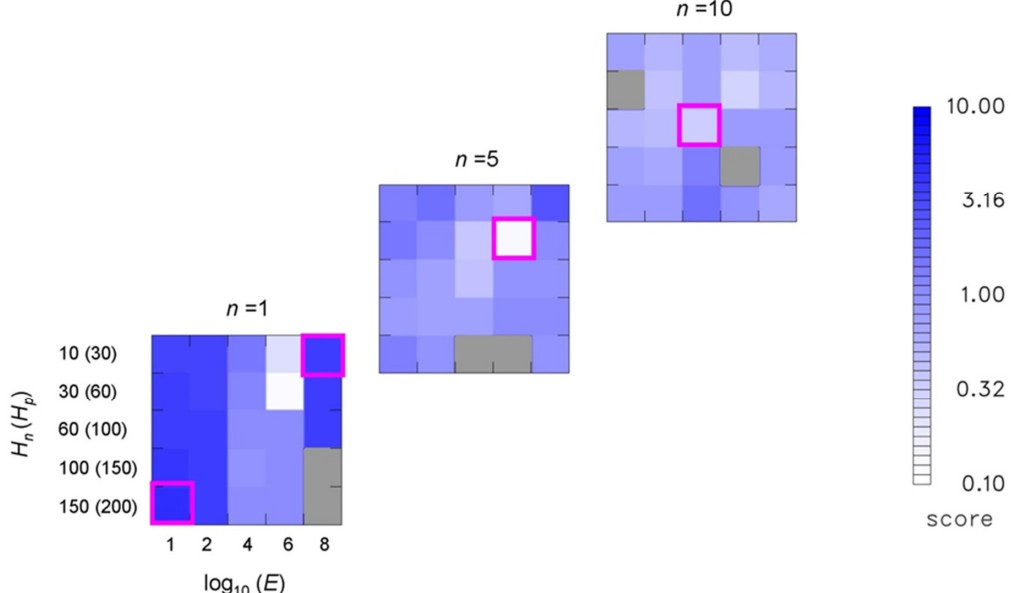

**Figure 4.** Scores for all runs in the ensemble of Jakobshavn simulations. Scores of 0 to ~0.3 indicate rough agreement with observed magnitudes of mélange thicknesses, velocities and extent, and scores of ~0.5 and greater indicate somewhat to very



unrealistic simulations (see Appendix C). Each square with 5-by-5 tiles shows scores for ranges of flow enhancement factor $E$ (horizontal axis, $10^1$ to $10^8$) and matched pairs of new-mélange and pressure-scaling thicknesses $H_n$ and $H_c$ (vertical axis, meters). The three squares show three values of the rheological coefficient $n$ (bottom left to top right, $n=1$, 5 and 10). Tiles towards the bottom left correspond to stiffer and thicker mélange, and those towards the top right correspond to weaker and

5    thinner mélange. Grey tiles indicate simulations that encountered numerical instability (see text). Magenta outlines identify the simulations shown in Fig. 3.

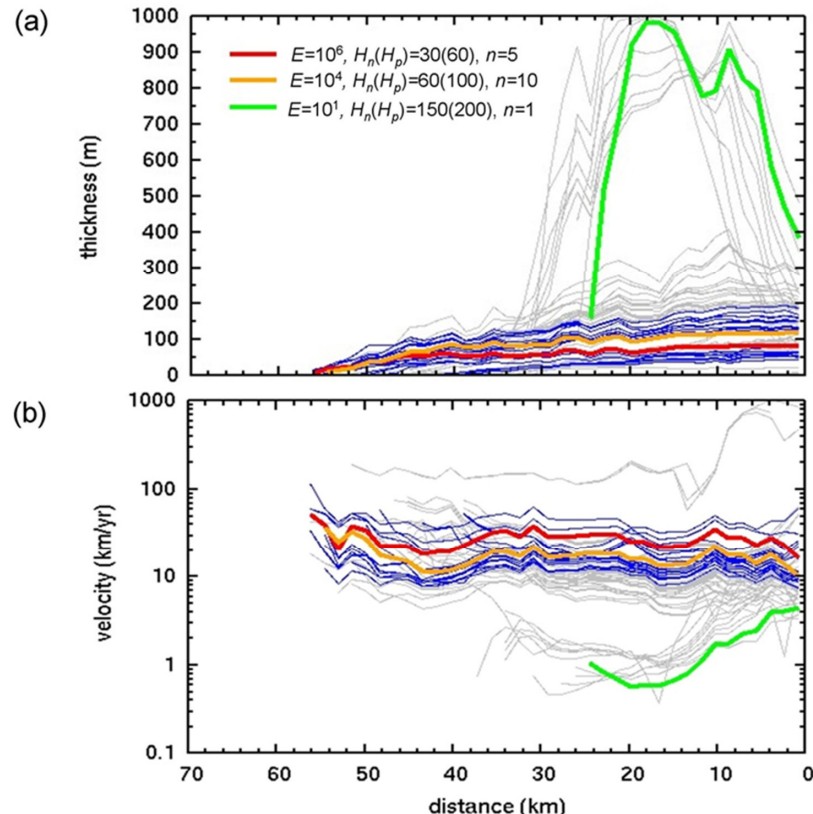

**Figure 5.** East-west profiles along the fjord for all simulations in the Jakobshavn ensemble. **(a)** mélange thickness (m). **(b)** mélange westward velocity (km yr$^{-1}$). Quantities are averaged over north-south transects across the fjord (as in the scoring calculations, Appendix C). The horizontal axis shows westward distance from the head of the fjord, defined as the most





eastward modeled grounding-line location. **Red:** near-best scoring run shown in Fig. 3a ($E=10^6$, $H_n(H_p)=30(60)$, $n=5$). **Orange:** another realistic run shown in Fig. 3b ($E=10^4$, $H_n(H_p)=60(100)$, $n=10$). **Green:** much stiffer and thicker mélange, shown in Fig. 3c ($E=10^1$, $H_n(H_p)=150(200)$, $n=1$). **Blue:** runs with the 20 best scores. **Grey:** all other runs.

### 4.3 Antarctica

We now use the coupled model to examine the role of mélange during rapid retreat of Antarctic ice. Starting from the ice-sheet model state equilibrated to modern climate (with no mélange), an instantaneous change to a warm ~3 Ma mid Pliocene climate is imposed. As described in Pollard et al. (2015), atmospheric forcing is provided by a regional climate model with a warm austral-summer orbit and atmospheric $CO_2$ level of 400 ppm, and circum-Antarctic ocean temperatures are assumed to warm 2 °C above modern climatology. The inclusion of hydrofracturing and cliff-failure mechanisms in the ice model produces very rapid West Antarctic ice sheet (WAIS) retreat within ~200 years, and major East Antarctic (EAIS) retreat in the Wilkes, Aurora and Recovery marine basins within ~3000 years (Pollard et al., 2015). Similar retreat occurs in future model simulations with the IPCC business-as-usual RCP8.5 greenhouse-gas scenario (DeConto and Pollard, 2016). The same ice model version as in those studies is used, except for (i) no lapse-rate adjustment to precipitation for the difference between climate-model and ice-model surface elevations, and (ii) an increase of maximum cliff erosional rate from 3 to 12 km yr$^{-1}$ (more like Jakovshavn today), both of which tend to increase marine ice retreat.

As mentioned above, the very fast mélange speeds in some regions and times (up to ~2000 km yr$^{-1}$) require very short timesteps ($\Delta t < \Delta x/2000$ for numerical stability), and long-term Antarctic runs are only practical at 40 km spatial resolution. Shorter regional tests at higher resolutions (with one example shown below) and the idealized tests in Fig. 2 suggest that the results are reasonably independent of model resolution.




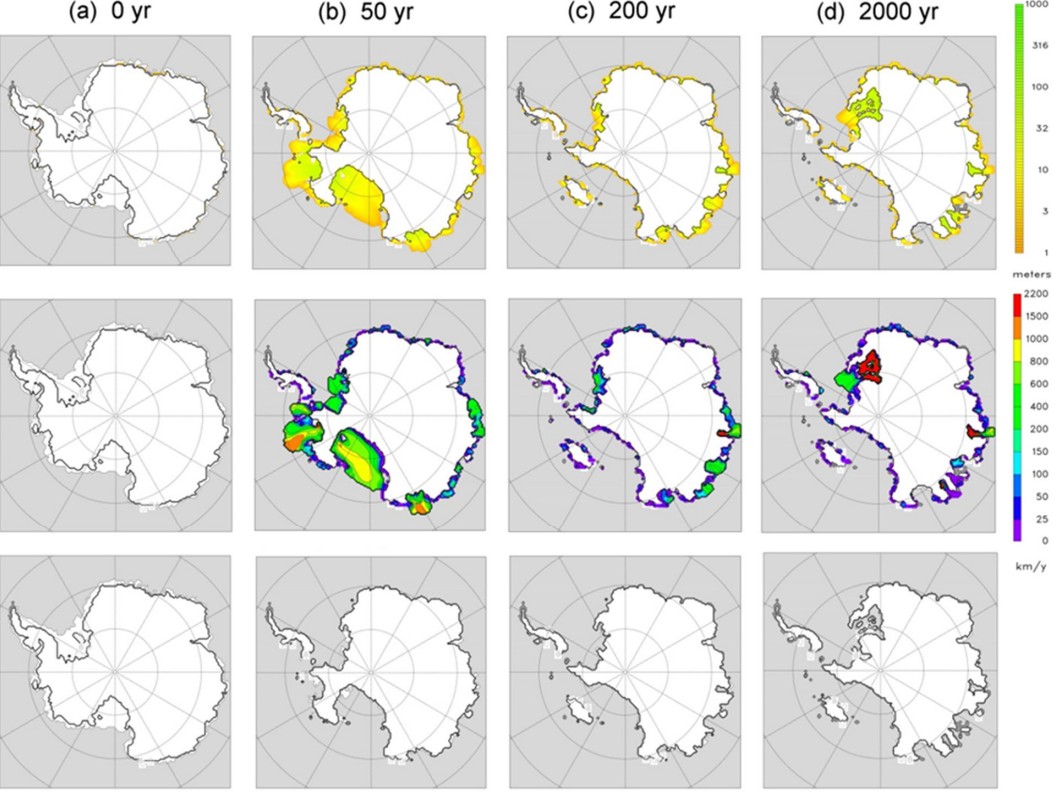

**Figure 6.** Snapshots of mélange thickness, m (**upper row**) and speed, km yr$^{-1}$ (**middle row**) in a simulation of Antarctic ice retreat, at selected times after a step-function transition from modern to warm mid-Pliocene climate. White regions are solid ice sheet or shelf, grey is ocean, and the grounding line is shown by a black line. Mélange parameter values are as in the near-best scoring run of the Jakobshavn ensemble shown in Fig. 3a ($E=10^6$, $H_n(H_p)=30(60)$, $n=5$). **Bottom row** shows solid ice extents from a corresponding simulation with no mélange. **(a)** 0 yr (modern). **(b)** 50 yr. **(c)** 200 yr. **(d)** 2000 yr.

Fig. 6 shows snapshots at selected times after the imposition of Pliocene climate, with mélange parameters set to the near-best scoring values in the Jakobshavn ensemble shown in Figs. 3a and 4 above. Large amounts of mélange are generated within 50 years by retreating ice in marine West Antarctica, producing mélange thicknesses up to 50 m in much of Amundsen and Ross embayments, with lesser thicknesses (~10 m) in the Weddell. However, the additional back stress of mélange on the ice faces has a negligible effect on WAIS collapse, which occurs almost at the same pace as in the model



without mélange (Pollard et al., 2015; Fig. 6 bottom row), and retreat of WAIS marine margins is nearly complete within ~200 years.

The same is true for later retreat into the major Wilkes, Aurora and Recovery basins of East Antarctica. Despite their more confined and shallow sills, mélange makes very little difference, and retreat into these basins occurs within ~2000

5 years as in the model with no mélange. Even with much stiffer and thicker mélange parameters (the combination used in Fig. 3c for Jakobshavn, far outside the realistic range), most of the retreat is largely unaffected, as seen in Fig. 7. West Antarctica and the Wilkes basin still collapse, and retreat is slowed or prevented only in some East Antarctic inlets, notably the Recovery basin east of the Weddell embayment.

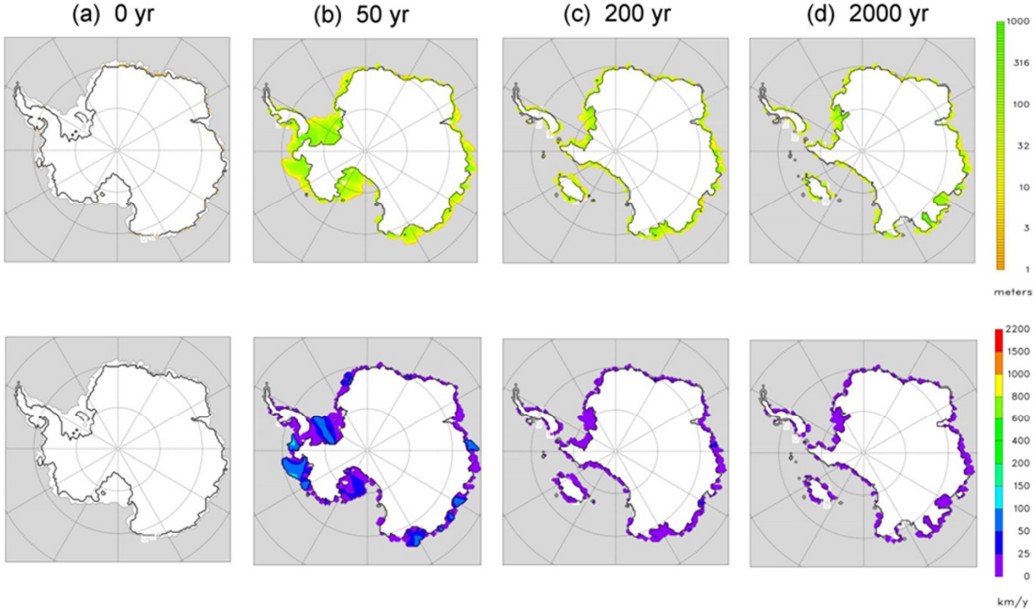

**Figure 7.** As Fig. 6 for parameter values producing very stiff and thick mélange as in Fig. 3c ($E=10^1$, $H_n(H_p)=150(200)$, $n=1$). **(a)** 0 yr (modern). **(b)** 50 yr. **(c)** 200 yr. **(d)** 2000 yr.

Corresponding time series of various quantities are shown for the two cases in Fig. 8. The equivalent global mean sea-

15 level (GMSL) for the near-best scoring parameter values (red curve in Fig. 8a) is nearly the same as in the model with no mélange. This is true for several other (we suspect all) combinations of parameters yielding reasonable mélange magnitudes for Jakobshavn in Fig. 4, for which total GMSL rise in these Antarctic simulations remains close to ~13 m. Note again that the other case with much stiffer mélange (green curve in Fig. 8a) yields very unrealistic results for Jakobshavn. Other





mélange quantities are shown in Fig. 8b-e, and vary as expected between the two cases. The pronounced rise in total additional back force with very stiff mélange after ~2200 years (green curve in Fig. 8e) is due to the delayed retreat into a single East Antarctic inlet around 130 °E, that has collapsed well before 2000 years with the more realistic mélange parameters (seen in Fig. 6d v. 7d).

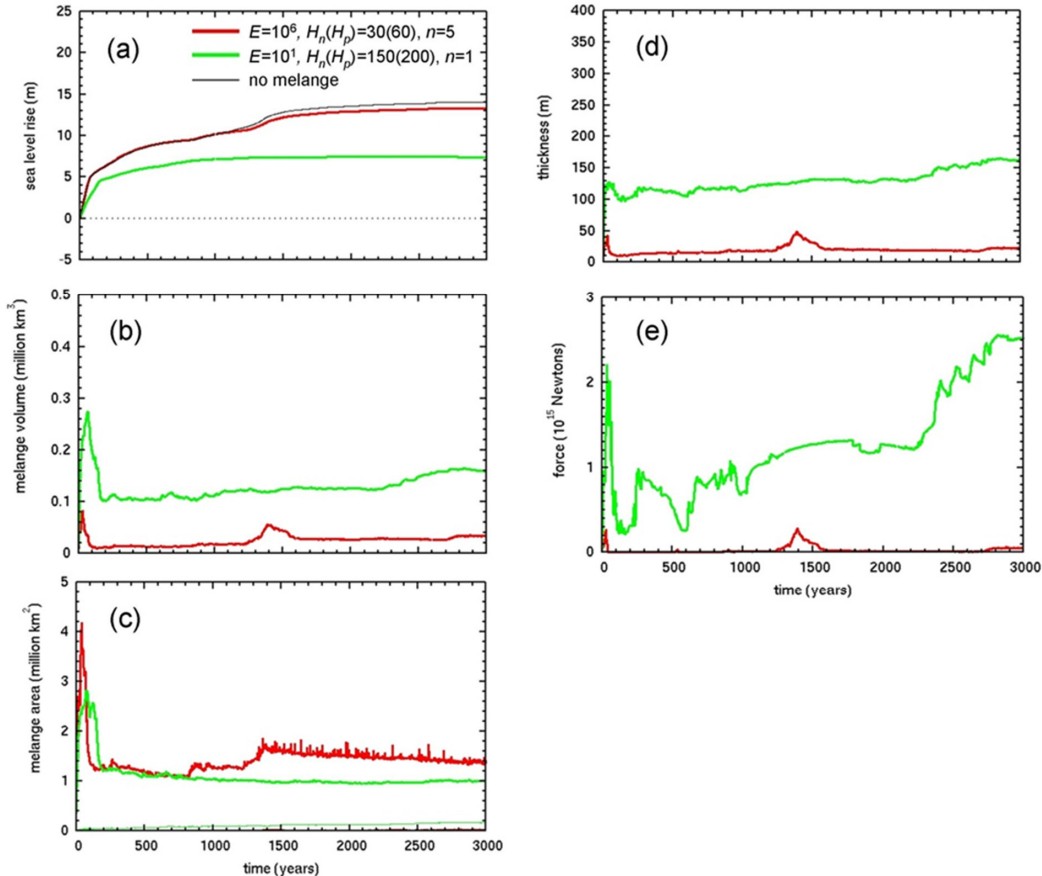

**Figure 8.** Time series in simulations of Antarctic retreat after a transition to warm mid Pliocene climate with two different mélange parameter settings. **Red curves:** $E=10^1$, $H_n(H_p)=150(200)$, $n=5$ (near-best scoring, as in Fig. 6). **Green curves:** $E=10^1$, $H_n(H_p)=150(200)$, $n=1$ (much stiffer and thicker mélange, as in Fig. 7). **(a)** Equivalent global mean sea level rise (m),

10 with black curve for a corresponding simulation with no mélange. **(b)** Total mélange volume ($10^6$ km$^3$). **(c)** Total mélange area ($10^6$ km$^2$), with thinner curves showing area where grounded. **(d)** Mélange thickness (m) averaged over all mélange grid



cells adjacent to a solid ice face. **(e)** Total additional back force due to mélange summed over all Antarctic ice faces, compared to that due to ocean water pressure with no mélange ($10^{15}$ Newtons; see Appendix A).

A nested run over the Wilkes basin was performed as a basic test of model resolution, shown in Fig. 9. This run

5  corresponds in all respects to the Pliocene retreat scenario in Fig. 6 above, except with the grid size reduced to 20 km, and with lateral boundary conditions at the domain margins fixed to modern ice. The distribution of mélange is slightly different, but very much the same grounding-line retreat into the Wilkes basin occurs as in the continental run.

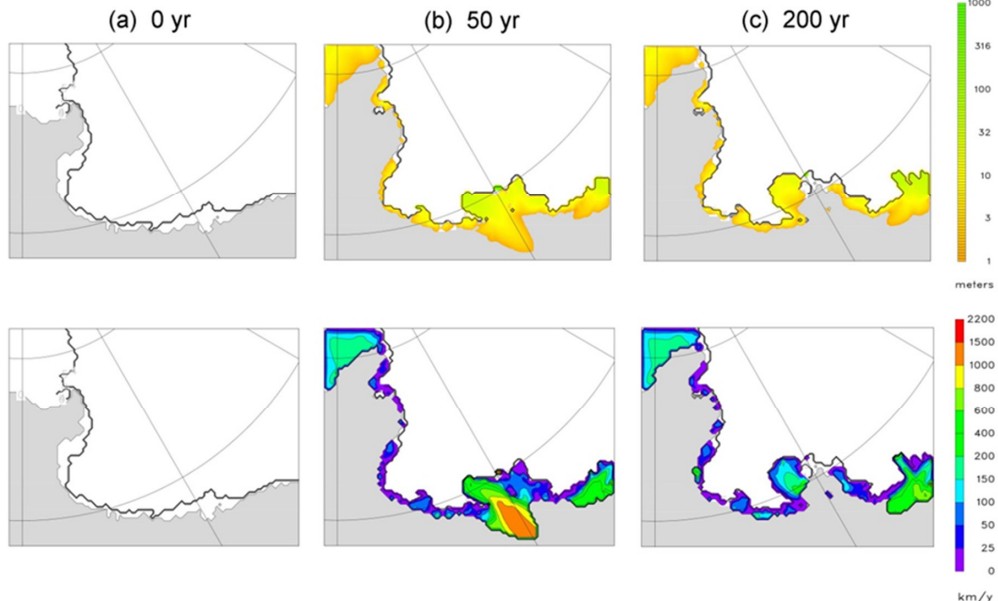

**Figure 9.** Snapshots of mélange thickness, m **(upper row)** and speed, km yr$^{-1}$ **(lower row)** in a limited-area nested

10  simulation at 20 km resolution over the Wilkes subglacial basin, East Antarctica, after a step-function transition to warm mid-Pliocene climate. Model parameters are as in Fig. 6, including mélange settings (near-best scoring, $E=10^6$, $H_n(H_p)=30(60)$, $n=5$). **(a)** 0 yr (modern). **(b)** 50 yr. **(c)** 200 yr.



## 5 Conclusions

A continuum-mechanical model of mélange has been formulated, that is computationally feasible for continental spatial and multi-millennial time scales. In idealized channel tests it captures basic dependencies between supply rate, flow, side drag and ocean bottom resistance, and their influence on back pressure on ice faces. The model behaves consistently over a wide range of grid sizes.

The model was tested in simulations of mélange in the Jakobshavn fjord, West Greenland, aiming to calibrate the main uncertain model parameters for viscosity, side drag and bedrock drag. Ranges of these parameters are found that yield roughly correct magnitudes of mélange thickness, velocity and extent; values outside these ranges yield very unrealistic results with excessive thicknesses (for stiffer values), or excessive speeds (for weaker values).

When applied to rapid Antarctic retreat events using warm mid-Pliocene climate as an example, the inclusion of mélange has little effect on the response of Antarctic ice, for parameter values that yield reasonable Jakobshavn simulations. Extensive ~50-m thick mélange covers the Amundsen and Ross embayments in the early stages, with thinner mélange in the Weddell, but the additional back pressure on the retreating ice faces does not slow down the retreat noticeably, and marine WAIS collapse still occurs within ~200 years. The same is true for later retreat into the major marine basins of East Antarctica, despite their narrower and shallower sills. The lack of influence of mélange is presumably a consequence of the huge spatial scales of Antarctic ice fronts and seaways compared to Greenland fjords.

Mélange also has negligible effect on West Antarctic retreat if the mechanisms of hydrofracturing and cliff failure are removed. In that case WAIS collapse still occurs due to increased oceanic sub-ice-shelf melt and Marine Ice Sheet Instability, but more slowly on ~thousand-year timescales, as in other models (Feldman and Levermann, 2015; Golledge et al., 2015; Ritz et al., 2015). Because of the slower retreat of ice margins, the generation of mélange is outpaced by removal to the open ocean with much less accumulation at the margins, and the effect on overall WAIS collapse is minimal (not shown). To produce significant slowdown of Antarctic retreat, mélange parameters must be set to much "stiffer" values, far outside the reasonable ranges for Jakobshavn. Even then, retreat is slowed or prevented only for some East Antarctic basins (notably the relatively restricted Recovery), and still occurs in West Antarctica and the Wilkes basin.

Within the framework of this study, the results strongly indicate that mélange has very little influence on major Antarctic ice retreat. However, the model clearly has large uncertainties. Even though it produces reasonable magnitudes of some mélange properties in Jakobshavn, that does not mean that the dynamical processes in the model accurately represent mélange there. Even if it does, its applicability to the much larger scales of Antarctica would still be questionable. One primary need is detailed mechanistic models of calving and cliff failure, to replace the simple empirical parameterizations currently used. Mélange may have greater effects in more mechanistic models of calving, for instance in preventing overturning of icebergs (Amundson et al., 2010, their Fig. 9, where the required mélange forces, $O(10^7$ to $10^8)$ N m$^{-1}$, are comparable to those in our Jakobshavn simulations). Another hopeful goal is to link the results of discrete-particle models of mélange with continuum descriptions as here. Beyond that, the following steps are planned for future work.





- Using higher resolution in Jakobshavn simulations, and performing more detailed and thorough calibration to modern observations. This would include seasonal variations due to winter freezing and hardening of the mélange by the sea-ice matrix, which prevents calving during winter (Joughin et al., 2008, 2014b; Amundson et al., 2010).

- Simulating past variations of Jakobshavn ice and mélange extents, during the Holocene, last two centuries and decades (Csatho et al., 2008; Joughin et al., 2008; Young et al., 2011).

- Improving the numerical treatment of the ice-mélange junction when it migrates across multiple grid cells. As described in Appendix B, this is done simply but not rigorously in the current model, and would involve fractional grid coverage for mélange, which is not yet in the model as it is for ice shelves (although no detrimental behavior has been seen to date; cf. Albrecht et al., 2011).

- Distinguishing between supply of mélange vs. large tabular bergs in the ice model's calving parameterization (Pollard and DeConto, 2012). For modern Antarctica, model calving at the edges of the Ross and Weddell shelves produces a small amount of mélange, contrary to observations (but has very little effect on the model's modern state).

- Allowing depth-dependent oceanic melt rates, appropriate for thick mélange in marine settings with steep vertical temperature gradients.

- Including mélange transport by ocean currents, particularly the circum-Antarctic western boundary current that today advects most icebergs counter-clockwise around the Antarctic coast and then northwards off the eastern Antarctic Peninsula through Iceberg Alley (Weber et al., 2014). Similar routing could influence the huge amounts of mélange generated during rapid retreat episodes.

**Code and Output Availability**

The coupled ice sheet and mélange model code is available on request from the corresponding author (pollard@essc.psu.edu). That and selected model output will be available at Penn State's Data Commons, http://www.datacommons.psu.edu.

**Acknowledgements**

This work was supported by National Science Foundation grants OCE-1202632, GEO-1240507, ANT-1341394 and PLR-1443190.



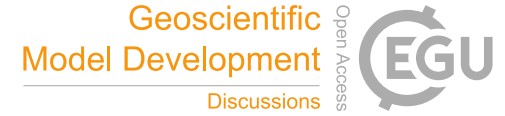

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



## Appendix A. Mélange back stress.

This appendix describes the force balance on ice shelves and mélange, as entities separate from grounded ice. The analysis is based on the net force on any of these bodies being essentially zero at any time. The next Appendix B describes

kinematic relationships involving the conversion of ice to mélange across the ice-mélange interface, based on conservation of mass. Both appendices derive equations that are used in the model as boundary conditions.

Back stress by mélange in contact with vertical solid-ice faces is computed in the model. This can occur either at the grounding line of marine ice cliffs with no ice shelf (tidewater termini), or at the outer edge of floating ice shelves. The back stress stems from the net forces on the mélange that resist downstream flow; i.e., side drag or blockage by land or ice, basal

drag if the mélange grounds on ocean bedrock rises, and (relatively small) friction between the mélange base and ocean water. The process is much the same as the well-known buttressing at grounding lines by ice shelves, but analysis is slightly more complicated because the mélange only occupies a part of the vertical ice face, and the junction is between two separate bodies, not within contiguous ice. Stretching forces are not transmitted across the boundary, and $\partial u/\partial x$ in the ice may be different from that in the mélange adjacent to the boundary (it is continuous across sheet-shelf grounding lines). The simpler

ice-shelf case is reviewed first in section A1, and the mélange case is analyzed in section A2, leading to the expression for mélange back stress used in the model.

The treatment below is for 1-D flowlines in the $x$ direction, with no transverse variation. We note below (after Eq. A2) how some expressions would be modified for transverse variability in the $y$ direction, still with the grounding line running perpendicular to the $x$ axis.

### A1. Ice-shelf back stress at a grounding line

Buttressing by an ice shelf of grounded ice flowing across the grounding line is accounted for in our existing ice-sheet model by the term $\theta$ in the equation for grounding-line ice velocity (Schoof, 2007, eq. 29; equal to $\tau_{xx}/\tau_f$; Pollard and DeConto, 2012, eq. 8). $\theta$ is the ratio of the longitudinal deviatoric stress at the grounding line, to its value if the ice shelf was

completely unconfined or did not exist. As in SSA scaling, vertical shear is neglected in the vicinity of and downstream of the grounding line, and all main quantities here are independent of height.

$$\theta = \left( 4\eta \frac{\partial u}{\partial x} \right) \bigg/ \left( 4\eta \frac{\partial u_f}{\partial x} \right) \tag{A1}$$

where the effective viscosity $\eta$ is given by Eq. 2 of the main text (with $n=3$). The reason for the factor 4 (which cancels in Eq. A1) is mentioned following Eq. A2 below.





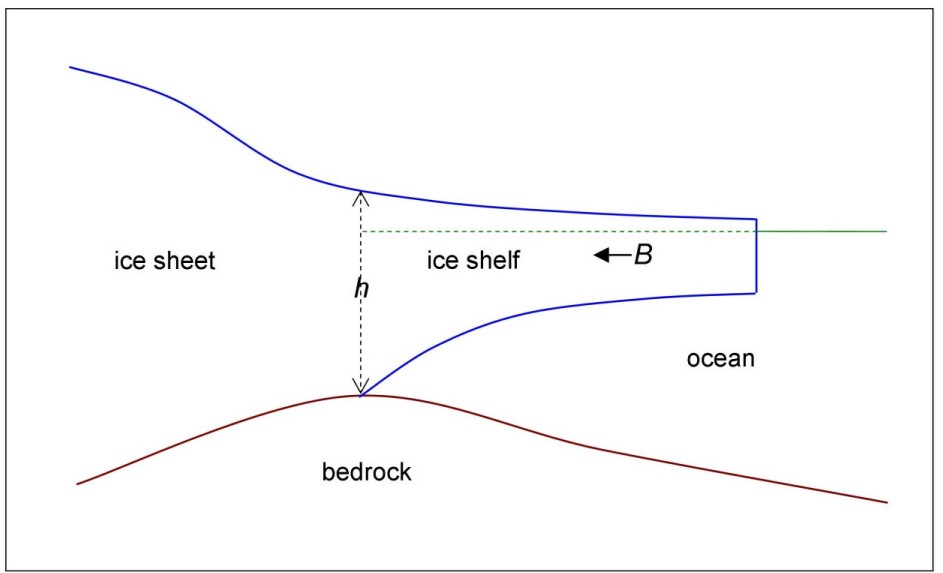

**Figure A1.** Schematic of an ice sheet and ice shelf at the grounding line. *B* is the net backward force on the ice shelf due to side drag or grounding on bedrock.

5  Referring to the schematic in Fig. A1, the net force (towards the right, vertically integrated, per unit length in the transverse direction) on the entire ice shelf must be zero, and is

$$-4\eta \frac{\partial u}{\partial x}h + \frac{\rho_i g h^2}{2} - \frac{\rho_w g}{2}\left(\frac{\rho_i}{\rho_w}h\right)^2 - B = 0$$

(A2)

Note that the extra factor of 2 in the $4\eta$ term at the start of Eq. A2, instead of $2\eta$, comes from the effect of vertical deviatoric stress $2\eta \ \partial w/\partial z$ in the vertical force balance with gravity (e.g., Thoma et al., 2014). For two horizontal dimensions (with the

10  grounding line still perpendicular to the *x* axis), the first term in (A2) would be $2\eta \ (\partial u/\partial x + (\partial u/\partial x + \partial v/\partial y)) \ h$, as in the SSA equations for velocity *u* (seen for mélange in Eq. 4a). For the two-dimensional case, wherever $4\eta \ \partial u/\partial x$ appears in this Appendix, it should be replaced by $2\eta \ (2\partial u/\partial x + \partial v/\partial y)$, including the subscripted terms $4\eta_i \ \partial u_i/\partial x$ and $4\eta_m \ \partial u_m/\partial x$ for ice and mélange separately in the next subsection.

If the shelf is freely floating (or non-existent), $B = 0$ in Eq. A2, and the unconfined strain rate $\partial u_f / \partial x$ is given by

$$-4\eta \frac{\partial u_f}{\partial x}h + \frac{\rho_i g h^2}{2} - \frac{\rho_w g}{2}\left(\frac{\rho_i}{\rho_w}h\right)^2 = 0$$

(A3)

Using Eq. (A1),





$$\theta = \left( 4\eta \frac{\partial u}{\partial x} h \right) \bigg/ \left( \frac{\rho_i g h^2}{2} - \frac{\rho_w g}{2} \left( \frac{\rho_i}{\rho_w} h \right)^2 \right)$$

i.e.,

$$\theta = \left( 4\eta \frac{\partial u}{\partial x} h \right) \bigg/ \left( \frac{\rho_i' g h^2}{2} \right) \tag{A4}$$

where $\rho_i' = (1 - \rho_i/\rho_w) \rho_i$. Eq. A4 is the expression used for $\theta$ in our ice sheet-shelf model, with $\partial u/\partial x$ obtained iteratively

from the model's velocity solution (Pollard and DeConto, 2012; Pollard et al., 2015; DeConto and Pollard, 2016).

The following relation, although not used in the model, clarifies the connection between $\theta$ and the net external forces $B$ resisting ice-shelf flow downstream (basal pinning points, side drag and/or blockage). Combining (A2) and (A4) yields

$$\theta = 1 - \frac{B}{\left( \rho_i' g h^2 / 2 \right)} \tag{A5}$$

So with $B = 0$, i.e., an unconfined or non-existent ice shelf, $\theta = 1$. As $B$ increases, $\theta$ decreases, and reaches 0 when $B = \rho_i'$

$g\ h^2/2$, i.e., when the combination of $B$ and total water pressure on the ice shelf exactly balances the column-integrated hydrostatic pressure at the grounding line, and stretching $\partial u/\partial x$ at the grounding line is zero (from Eq. A2).

**A2. Mélange back stress at a tidewater cliff or an ice-shelf edge**

The analysis of mélange back stress is similar to the above, but is more involved because (i) the mélange occupies only a

portion of the vertical ice face, and (ii) the junction is between two distinct bodies, so longitudinal stress is not transmitted across it.

Just as for ice-shelf buttressing above, $\theta_i$ is the ratio of longitudinal strain in the solid ice adjacent to the mélange, relative to what it would be with no mélange:

$$\theta_i = \left( 4\eta_i \frac{\partial u_i}{\partial x} \right) \bigg/ \left( 4\eta_i \frac{\partial u_{if}}{\partial x} \right) \tag{A6}$$

Unlike the ice-shelf case, this applies to solid ice immediately upstream of the face. The subscript $i$ indicates solid-ice quantities, and subscript $m$ below indicates mélange quantities. $\theta_i$ is used by the ice-sheet model to account for mélange back stress, just as $\theta$ from Eq. A4 is used for ice-shelf back stress. Note that throughout this section, the mélange viscosity $\eta_m$ includes the factor $f$ for mélange convergence or divergence, i.e., $\eta_m = f \eta$ where $\eta$ is given by Eq. 2.



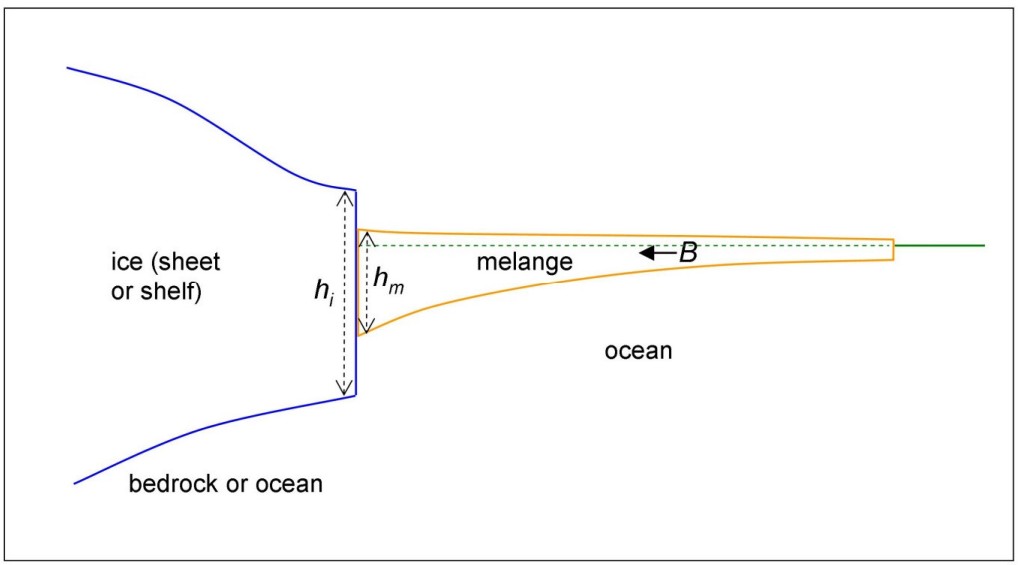

**Figure A2.** Schematic of mélange at the vertical face of an ice sheet or ice shelf. $B$ is the net backward force on the mélange due to side drag, grounding on bedrock, or blocking ice further downstream.

Referring to Fig. A2, the leftward force $F_m$ exerted by the mélange on the ice face (the portion of the ice face that it touches), equal to the rightward force exerted by the ice on the mélange, is

$$F_m = -4\eta_m \frac{\partial u_m}{\partial x} h_m + \frac{\rho_m g h_m^{\,2}}{2} + P_p$$

(A7)

$P_p$ is the vertically integrated pressure term representing packing given by Eq. 5. $\rho_m$ is the bulk density of mélange, which may be slightly greater than ice density $\rho_i$ due to embedded liquid within the mélange. As noted for ice after Eq. A2 above,

the first term on the right in (A7) with longitudinal stretching $\partial u_m/\partial x$ enters not because of any direct stretching or compression of the ice, but because of the modification to hydrostatic pressure in the mélange adjacent to the ice, via the effect of deviatoric stress $2\,\eta_m\,\partial w_m/\partial z$ in the vertical force balance.

The net force (towards the right) on the entire body of mélange must be zero, and is

$$F_m - \frac{\rho_w g}{2}\left(\frac{\rho_m}{\rho_w} h_m\right)^2 - B \;\; = \;\; -4\eta_m \frac{\partial u_m}{\partial x} h_m + \frac{\rho_m g h_m^{\,2}}{2} - \frac{\rho_w g}{2}\left(\frac{\rho_m}{\rho_w} h_m\right)^2 - B + \; P_p \; = \; 0$$

(A8)

which also follows by integrating Eq. 4a in the $x$ direction from an open-ocean boundary to the grounding line, using the open-ocean boundary condition Eq. 6, and $B$ representing the integrated basal friction. $B$ can also include side friction arising





from the second term on the left of Eq. 4a (involving $\partial U/\partial y$), if it is also integrated in the $y$ direction between walls running parallel to the $x$ axis.

Similarly, the net force (towards the left) on the solid ice face, due both to the mélange contact and the hydrostatic pressure of ocean water, is

$$F_i = F_m + \int_{\frac{\rho_m}{\rho_w}h_m}^{\frac{\rho_i}{\rho_w}h_i} \rho_w g\, z\, dz$$

(A9)

And this must be in balance with the rightward force on the vertical face in the ice immediately upstream, so $F_i$ also obeys

$$F_i = \frac{\rho_i g h_i^2}{2} - 4\eta_i \frac{\partial u_i}{\partial x} h_i$$
(A10)

Combining (A6) and (A10),

$$\theta_i = \frac{\left(\rho_i g h_i^2/2\right) - F_i}{\left(\rho_i g h_i^2/2\right) - F_{if}}$$
(A11)

Using (A7) and (A9) in (A11),

$$\theta_i = 1 - \frac{\left(\rho_m' g h_m^2/2 + P_p\right)}{\rho_i' g h_i^2/2}\left(1 - \frac{4\eta_m h_m\left(\partial u_m/\partial x\right)}{\left(\rho_m' g h_m^2/2 + P_p\right)}\right)$$
(A12)

where $\rho_m' = (1 - \rho_m/\rho_w)\,\rho_m$. and $\rho_i' = (1 - \rho_i/\rho_w)\,\rho_i$. This expression for $\theta_i$ is used in the model to account for back pressure by mélange at ice cliff faces (Pollard et al., 2015), and also to modify the boundary condition in the SSA dynamics at the edges of ice shelves (Pollard et al., 2012) (noting again that $\eta_m$ includes the factor $f$ for mélange convergence or divergence). Note

that Eq. A12 can be written

$$1 - \theta_i = \frac{\left(\rho_m' g h_m^2/2 + P_p\right)}{\rho_i' g h_i^2/2}\left(1 - \theta_m\right)$$

where

$$\theta_m = \frac{4\eta_m h_m\left(\partial u_m/\partial x\right)}{\left(\rho_m' g h_m^2/2 + P_p\right)}$$
(A13)

$\theta_m$ is the factor representing degree of buttressing in the mélange adjacent to the face ($\theta_m = 1$ if free flow, $\theta_m \leq 0$ if fully

buttressed), and corresponds to $\theta_i$ for ice.

As for the ice-shelf case above, additional relationships can be written that, although not used in the model, clarify the connection between $\theta_i$, $\theta_m$, and the net external forces $B$ resisting mélange flow downstream (bedrock grounding, side drag and/or blockage). Substituting (A8) and (A9) into (A11) leads to:



$$\theta_i = 1 - \frac{B}{\left(\rho_i' g h_i^2/2\right)} \tag{A14}$$

$$\theta_m = 1 - \frac{B}{\left(\rho_m' g h_m^2/2 + P_p\right)} \tag{A15}$$

This is exactly the same form as Eq. A5 for the ice-shelf case above (without $P_p$), and the same discussion applies for mélange. If $B$ and $P_p = 0$, i.e., no downstream external forces on the mélange (except water pressure), then $\theta_i = \theta_m = 1$, and

5 the ice face feels the same force as if no mélange exists. As $B$ increases, $\theta_m$ decreases and reaches 0 when $B = \rho_m' g h_m^2/2 + P_p$, i.e., when the combination of $B$ and total water pressure on the mélange exactly balances the column-integrated hydrostatic and packing pressure in the mélange adjacent to the solid ice face, and mélange stretching $\partial u_m/\partial x$ is zero. Note that at that point, $\theta_i$ is not necessarily zero. If $B$ is larger, $\theta_m$ becomes negative, i.e., the mélange is being compressed immediately downstream of the face with $\partial u_m/\partial x < 0$, which does occur in some of our simulations with large side drag or

10 grounded mélange.

Another quantity of interest is the net force (towards the left) on the solid ice face, due both to the mélange contact and the hydrostatic pressure of ocean water, minus what it would be just due to ocean water pressure in the absence of mélange, i.e, $F_i - F_{if}$. Rearranging Eq. A11, and using $F_{if} = \rho_w g (\rho_i/\rho_w)^2 h_i^2/2$ (i.e., A9 with $h_m = 0$), and also A13, yields

$$F_i - F_{if} = (1 - \theta_i)\frac{\rho_i' g h_i^2}{2} = (1 - \theta_m)\left(\frac{\rho_m' g h_m^2}{2} + P_p\right) \tag{A16}$$

This expression is used in the model to calculate the total force difference $F_i - F_{if}$ integrated over all ice faces in contact with mélange, as a diagnostic domain-wide measure of mélange buttressing during a simulation. Another quantity of interest is the leftward force on the ice face exerted by the mélange itself, $F_m$ in Eq. A7. Combining A7 and A13,

$$F_m = (1 - \theta_m)\left(\frac{\rho_m' g h_m^2}{2} + P_p\right) + \frac{\rho_m^2 g h_m^2}{2\rho_w} \tag{A17}$$

Similarly, the total leftward force on the ice face due to the mélange and water, $F_i$ in Eq. A9, is

$$F_i = (1 - \theta_m)\left(\frac{\rho_m' g h_m^2}{2} + P_p\right) + \frac{\rho_i^2 g h_i^2}{2\rho_w} \tag{A18}$$





**Appendix B. Ice-to-mélange conversion**

This appendix analyzes the relatively narrow zone where solid ice is converted to mélange, at a tidewater face of

5 grounded ice, or the open-ocean edge of an ice shelf. The zone is relatively narrow compared to the grid size, not resolved in the model, and the physics of calving or cliff failure that occur within it are not addressed here (their net rates are parameterized within the ice model). Relations based on conservation of mass are derived that relate the net effect of the conversion to model variables, mainly to provide the correct velocity boundary condition at the face for the mélange SSA velocity equations. Following that, issues with implementation in the model's finite-difference grid are described.

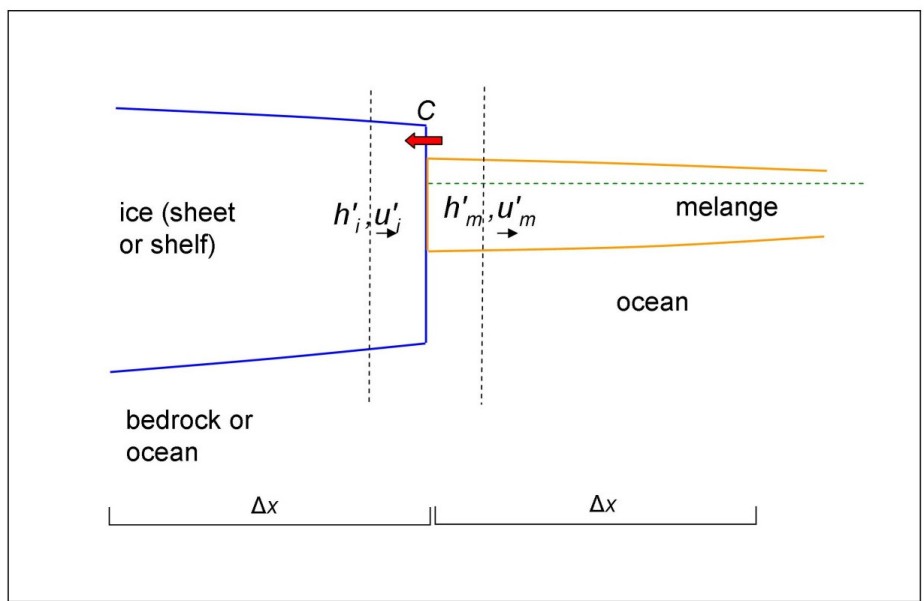

**Figure B1.** Schematic of the ice-mélange interface zone (between the two vertical dashed lines), in which new mélange is generated by calving or cliff failure of the ice face. $h_i'$ and $u_i'$ are the thickness and velocity, respectively, of solid ice just upstream of the ice face, and $h_m'$ and $u_m'$ are the same for mélange just downstream of the face. $C$ is the rate of cliff or calving erosion horizontally into the ice interior. Two model $h$-grid cells ($\Delta x$) are indicated at the bottom.

Referring to Fig. B1, $h_i'$ and $u_i'$ are the thickness and velocity, respectively, of solid ice just upstream of the ice face, and $h_m'$ and $u_m'$ are the same for mélange just downstream of the face. A prime ′ is used to indicate that these quantities are not the same as grid-center quantities in the finite-difference model, for which two grid-box extents ($\Delta x$) are sketched in the





figure. $u_i'$ and $u_m'$ are Eulerian velocities relative to the fixed grid. In contrast, $C$ is the rate of cliff or calving erosion horizontally into the ice interior, i.e., the rate at which ice is converted to mélange, in volume per time per unit lateral width and unit vertical face height, and is not Eulerian. $C$ plays a similar role in the description of velocities at calving faces with no mélange (Benn et al., 2007).

The two dashed vertical lines in Fig. B1 denote a narrow interface zone in which calving or cliff failure process occurs, not governed by the SSA equations. This zone is assumed to be narrow compared to the grid size. By conservation of mass between the two dashed lines in Fig. B1 (which are fixed relative to the grid, and for a short enough time interval that the interface zone remains between the lines):

$$s\left(\left|u_i'\right| - C\right)\left(\rho_i h_i' - \rho_m h_m'\right) = \rho_i\, h_i'\, u_i' - \rho_m h_m'\, u_m'$$
(B1)

The term $s(|u_i'| - C)$ is the Eulerian velocity of the interface, where $s = +1$ as drawn or $-1$ if reversed (mélange on the left, $u'$'s negative). Multiplication by the difference in ice and mélange thicknesses at the interface gives the rate of total mass increase between the dashed lines, which must be balanced by the fluxes across the lines. Rearranging yields

$$u_m' = u_i' + sC\left(\frac{\rho_i h_i'}{\rho_m h_m'} - 1\right)$$
(B2)

Note that if the interface is stationary ($C = |u_i'|$), Eq. B2 implies $\rho_m\, h_m'\, u_m' = \rho_i\, h_i'\, u_i'$, as required for overall mass
conservation. Also, if there is no calving or cliff failure ($C = 0$), $u_m' = u_i'$, i.e., the mélange is simply pushed down the fjord by the advancing ice. And if $\rho_m\, h_m' = \rho_i\, h_i'$, movement of the interface has no effect on mass distribution, and both of these statements are true.

$u_m'$ given by (B2) is the mélange velocity at the interface, needed in the mélange model as the boundary condition for Eq. 4a at ice interfaces perpendicular to the $x$ axis (and similarly $v_m'$ for the other dimension). Note that the use of (B2) captures
the "push-pump" mechanism mentioned in section 2, where the wintertime advance of the Jakobshavn ice face pushes mélange downstream, and its summertime retreat allows the space to be occupied by freshly created mélange. Here the net annual effect is captured with annual mean values of $u_i'$ and $C$ (which are close to equal for modern Jakobshavn's stationary ice front), and the entire mélange can be pushed down the length of the fjord via Eq. B2.

*Finite difference considerations:*

As long as the ice-mélange interface zone remains within a grid box and does not migrate across grid divisions, the ice-face boundary condition Eq. B2 is implemented easily and naturally. All quantities on the right-hand side of (B2) are known within the physics of the ice model, except for erosion rate $C$ and the thickness of new mélange $h_m'$ created immediately below the ice face (specified below). Because model velocity and thickness grids are staggered by half a grid box, the
velocities in Fig. B1 at the face are natural grid velocities at the interface between the two $h$-grid boxes sketched in the figure, and $u_i'$ is readily available. The ice thickness $h_i'$ can either be set to the upstream grid quantity $h_i$, or by the sub-grid





grounding-line interpolation already in the ice model (both have pluses and minuses, but yield very little difference in results).

$C$ and $h_m'$ depend on the physics of calving and/or cliff failure inside the interface zone, not on the SSA scaled physics outside. $C$ is already parameterized in the ice model (empirically using observed calving and cliff erosion rates, Pollard et al.,

2015). $h_m'$ is the thickness of newly created mélange immediately adjacent to the ice face, and is a new quantity that must also be parameterized (absent a detailed treatment of calving/cliff mechanics). Here, we simply set

$$h_m' = \min\left[ H_n, \frac{\rho_i}{\rho_m} h_i' \right]$$

(B3)

$H_n$ is a constant thickness of newly created mélange. Right at the ice faces of Jakobshavn and other Greenland glaciers it is observed to be ~30 to 100 m thick. In the runs in this paper it ranges from 10 m to 150 m, and is matched to $H_p$ in Eq. 5, the

scaling value above which internal pressure increases rapidly due to packing, so that $H_p$ exceeds $H_n$ by ~20 to 50 m (values are given for each run above). Clearly, these parameterizations are crude and somewhat ad-hoc, and more study is needed on the use of Eqs. 5 and B3, and on appropriate values of $H_n$ and $H_p$ for Greenland and Antarctica.

If the ice-mélange interface advances or retreats across multiple grid cells, additional finite-difference steps are necessary. The procedure in the current model is simple, and makes the mélange a "slave" to the ice. The ice model already

has parameterizations for sub-grid fraction of ice shelves and grounding-line position (Pollard and DeConto, 2012), but the mélange model does not yet, and mélange coexists with ice in cells with fractional ice. If the ice-mélange interface advances across grid cells, the displaced mélange in these cells is immediately redistributed into adjacent mélange cells, conserving mélange and ice mass. If the interface retreats across grid cells into the ice interior, mélange already exists in the vacated cells, which it shared with partial cover of floating ice shelf before the adjacent grounded ice retreated, again conserving

mélange and ice mass.





**Appendix C. Scoring for Jakobshavn ensemble.**

A single score is computed for each simulation of modern Jakobshavn conditions, in the ensemble of runs described in section 4.2. Given the lack of 2-D maps of "climatological" annual mean mélange properties for Jakobshavn, the simulations

are scored simply versus ranges of fjord-wide average values with no spatial or seasonal dependence. These are estimated roughly from previous studies on mélange in Jakobshavn and other Greenland fjords. For mélange thickness, we use a range of 50 to 150 m based on freeboard elevations (Enderlin et al., 2016). For down-fjord velocities, we use a range of 20,000 to 60,000 m yr$^{-1}$ based on various data for Greenland fjords (Sundal et al., 2013; Foga et al., 2014; Amundson et al., 2016) and residence times (Enderlin et al., 2016), roughly accounting for strong seasonal variations in velocity and when the data were

taken (slower in winter corresponding to ice flow, faster in summer). For total mélange length, we use a generous range of 40,000 to 70,000 m, since the total length in the model is influenced strongly by the balance between supply rate at the ice face and oceanic basal melt, which may be inaccurate and are not part of the mélange model itself.

At the end of each simulation, 1-D profiles of mélange thickness and east-west velocity are generated by averaging quantities in north-south transects across the fjord wherever mélange exists, yielding $\overline{h_m}(x)$ and $\overline{u_m}(x)$ where $x$ is east-west

distance along the fjord. These values are penalized where they are outside the above ranges, according to

$$S_h(x) = \max\left(\log_e(\overline{h_m}/150), 0\right) + \max\left(\log_e(50/\overline{h_m}), 0\right)$$  (C1a)

$$S_u(x) = \max\left(\log_e(\overline{u_m}/60000), 0\right) + \max\left(\log_e(20000/\overline{u_m}), 0\right)$$  (C1b)

$$S_l = \max\left(\log_e(L_m/70000), 0\right) + \max\left(\log_e(40000/L_m), 0\right)$$  (C1c)

where $L_m$ is the total east-west extent of model mélange (truncated at the mouth if any mélange exists westward of 51.0º W).

The final score $S$ is the sum of the east-west averages of $S_h$ and $S_u$, and $S_l$

$$S = \int\left(S_h(x) + S_u(x)\right).dx \Big/ L_m + S_l$$  (C2)

Thus, thicknesses, velocities and total length incur penalties where they are larger or smaller than the acceptable ranges, with $O(1)$ (or larger) penalties for errors on the same (or larger) order as the ranges themselves. Smaller (larger) values of $S$ indicate simulations that are closer to (further from) the acceptable ranges. A model simulation with $\overline{h_m}(x)$ and $\overline{u_m}(x)$ (for

all $x$) and $L_m$ within the acceptable ranges would have $S = 0$.

The resulting scores for the ensemble are shown in Fig. 4. Given the preliminary nature of this study, the intent is just to identify which runs are in rough agreement with general magnitudes observed in Jakobshavn today (score values of 0 to ~0.3), versus runs that are somewhat to wildly unrealistic (score values of ~0.5 or more). A full-blown study of ice and mélange in Jakobshavn fjord would clearly require more detailed and comprehensive comparisons with data, both spatially

and temporally, as mentioned in the conclusions.