# Peer review of "A continuum model (PSUMEL1) of ice mélange and its role during retreat of the Antarctic Ice Sheet"

_Geoscientific Model Development, 2018_

## Short Comment (SC1) · 20 Mar 2018

Dear authors,

in my role as Executive editor of GMD, I would like to bring to your attention our Editorial version 1.1: http://www.geosci-model-dev.net/8/3487/2015/gmd-8-3487-2015.html This highlights some requirements of papers published in GMD, which is also available on the GMD website in the 'Manuscript Types' section: http://www.geoscientific-model-development.net/submission/manuscript_types.html In particular, please note that for your paper, the following requirement has not been met in the Discussions paper:

- "The main paper must give the model name and version number (or other unique

identifier) in the title."

Please provide the name and the version number of the ice melange model in the title of your revised manuscript.

As explained in https://www.geoscientific-model-development.net/about/manuscript_types.html GMD is encouraging that authors to upload the program code of models (including relevant data sets) as supplement or make the code and data of the exact model version described in the paper accessible through a DOI (digital object identifier). In case your institution does not provide the possibility to make electronic data accessible through a DOI you may consider other providers (eg. zenodo.org of CERN) to create a DOI. Please note that in the code accessibility section you can still point the reader how to obtain the newest version. If for some reason the code and/or data cannot be made available in this form (e.g. only via e-mail contact) the "Code Availability" section need to clearly state the reasons for why access is restricted (e.g. licensing reasons).

Yours, Astrid Kerkweg

---

## Referee Comment (RC1) · J.M. Amundson (Referee) · 2 May 2018

**Overview**

In this study the authors modify a set of equations used to model ice shelves in order to model the flow of ice mélange and to test its impacts on the behavior of the Antarctic Ice Sheet. Over the past decade or so, several observational papers have highlighted the relationship between ice mélange mobility and the seasonal advance and retreat cycle of marine-terminating glaciers. Few studies have attempted to model ice mélange, and only a couple have attempted to model ice mélange using continuum models. Development of continuum models of ice mélange is necessary for implementation into prognostic glacier/ice sheet simulations that commonly run for centuries to millenia. Thus, this paper is an important step toward implementing ice mélange into climate

models.

My primary concerns revolve around the choice of the constitutive relationship for ice mélange. Essentially, the authors use Glen's Flow Law, which describes the viscous deformation of glacier ice, to model the motion of a granular material. The modifications that they make are based on knowledge of granular materials (little/no resistance to divergence) and sea ice (compression becomes difficult when the thickness becomes large). It's possible that these modifications are sufficient to yield a good approximation of ice mélange behavior, but I'm not sure. Unfortunately, there is very little data with which to test *any* constitutive relation for ice mélange, and so the approach adopted here seems about as good as any other, especially since the architecture is already in place for solving the flow equations. That said, the results from the model need to be interpreted with caution, which I think the authors have done.

An additional uncertainty in the model is the lack of seasonality. In the model ice mélange generally exhibits extensional flow due to glaciostatic pressure gradients (similar to ice shelves). However, ice mélange flow in winter is often quasi-static, with all icebergs moving at the same speed as the glacier terminus. (In other words, pressure gradients are completely balanced by shear stresses along the fjord walls.) Ultimately, what this means is that the model equations used here probably underestimate ice mélange resistance in winter and overestimate it in summer. I would like to be convinced that this produces similar results to a model in which the ice mélange is really stiff in winter and weak in summer. On a related note, I'm not entirely sure if the boundary condition at the end of the mélange is correct, since it implies that gravitational spreading should occur even when the ice is very thin. Sea ice models are generally able to ignore this effect since sea ice typically moves very quickly; however, in models of landfast sea ice (e.g., Leppäranta, 2012) the constitutive relation is modified to ensure that gravitational spreading doesn't occur for thin ice by including a pressure term similar to the $P_p$ term that the authors use to inhibit compression of

thick mélange. It may be that the boundary condition is fine since the authors aren't modeling winter mélange and are just trying to capture the net annual effect. I'm just not really sure how to think the boundary condition in this case.

**Comments**

- Please replace the reference to Amundson et al. (2016) with Burton et al. (2018). The former is not a peer-reviewed article, and the latter includes the same material (and more). A notable result of Burton et al., which is relevant here, is that ice mélange resistance is related exponentially to the ice mélange length-to-width ratio ($L/W$). (Note that we have also found similar results using a continuum approach and assuming a Coulomb-like rheology, though that work is currently in review.) Assuming parallel-sided fjords and ignoring any potential binding effects of sea ice, we found that ice mélange was only capable of inhibiting calving when $L/W > 3$. This is consistent with the results from this study, which indicate the ice mélange doesn't have much effect on the wide ice shelves found in Antarctica. It could be interesting to compare — what is the length-to-width ratio of fjords in Antarctica for which the model predicts that ice mélange will affect the rate of retreat?

- I think the modifications to the constitutive relationship could be made more compact and transparent. I think you could write something like this:

$$\tau' = 2\eta\dot{\epsilon}_{ij}(1 + (f-1)\delta_{ij}) - P_p\delta_{ij} \tag{1}$$

As it currently stands, the reader has to get through some other details before reading about these modifications, which to me are really the most important parts of the paper. $P_p$ kind of appears out of nowhere, since it wouldn't naturally arise from vertical integration of the stress balance equations unless it appears

in the constitutive relation, and the description of $f$ is buried several paragraphs after it is first introduced.

By formulating the constitutive relation similarly to how I have here, you can immediately point out that (i) the equation is identical to the standard equations for glacier flow when $i \neq j$, (ii) there is low resistance to extension when $i = j$, and (iii) the $P_p$ term prevents $\dot{\epsilon}_{xx}$ or $\dot{\epsilon}_{yy}$ from becoming highly negative (compressive) when the mélange is thick.

**Detailed Comments**

P. 1, L. 10: Its not clear that ice melange noticeably affects glacier velocities directly.

P. 2, L. 9: This statement makes it sound like ice mélange is often just icebergs connected by sea ice, and not a densely-packed granular material.

P. 3, L. 23: Also probably affected by currents, wind, and tides (which would have an asymmetric influence).

P. 4, L. 5: Should be Vankova and Holland.

P. 4, L. 25: I agree that large icebergs are never overriden, but there could be some small scale rotation occurring, resulting in storage of gravitational potential energy.

**References**

Burton, J.C., J.M. Amundson, R. Cassotto, C.-C. Kuo, and M. Dennin (2018), Quantifying flow and stress in ice mélange, the world's largest granular material. Proc. Nat. Acad. Sci., https://doi.org/10.1073/pnas.1715136115

Leppäranta, M. (2012), The drift of sea ice, 2 ed. Springer-Verlag, Berlin and Heidelberg.

---

## Referee Comment (RC2) · N.R. Golledge (Referee) · 3 May 2018

This paper presents a new and innovative approach to the simulation of ice melange as an integrated component of a continental-scale ice sheet model. The subject is topical, and such developments are necessary for ensuring that future-focused Greenland and Antarctic ice sheet simulations capture processes relevant to their likely evolution, particularly with regard to how each ice mass might contribute to changes in future sea level. The basic approach of the paper is to use and modify equations of the shallow shelf approximation to simulate the ice melange as a thin and poorly-aggregated 'ice shelf' that exerts some back force against the calving face of the true ice shelf. Taking a continuum approach rather than employing fracture mechanics enables the scheme to integrate more easily with the ice sheet / shelf model equations.

[Figure]

The paper is very well-written, methodical, and clear, it is well illustrated and the arguments are easy to follow. My comments are primarily suggestions that could better define the effects of the melange, which would implicitly address some of the 'uncertainty' in simulating this kind of poorly-observed process.

Fundamentally my greatest concern is that there is a significant disconnect between the training scenario - a narrow Greenlandic fjord under present conditions - and the test scenario - the entire Antarctic ice sheet under a drastically warmer than present climate. Because of this, I feel that the Antarctic results, and the conclusions that depend on them, are not sufficiently convincing. By considering only two scenarios, the reader is left with no sense of any thresholds or dependencies in the system, which I feel are essential in terms of process understanding. I would like to see the following experiments added to help demonstrate the relative importance of the various model components in dictating how effective melange backpressure actually is:

1) Rather than just one 'Pliocene' scenario, why not use a range of step warmings in both atmosphere and ocean, with and without the melange feedbacks, to define the point at which melange becomes irrelevant? These should start at modern conditions and increment gradually, because at the moment the imposed 2deg C ocean warming is so high that it will almost certainly lead to rapid melt of the melange before any buttressing can exert an influence on GL location.

2) Following from that, there needs to be better separation of the model components, so a set of duplicate experiments are necessary in which, for a small or modest warming scenario, the effects of removing each of the important model components can be seen - i.e. - for a given ocean warming, show how GL retreat differs when a) ocean melt of the melange is turned off, b) shelf hydrofracture is turned off, c) tidewater cliff collapse is turned off. These kind of sensitivity experiments are alluded to in the conclusions, but not shown, which I think is a shame.

If these experiments all show that the inclusion of melange processes has no effect

on GL retreat, then the conclusions of the paper will be a lot more robust, and the modelling community will be happy that we don't need to add such schemes to existing models.

---

## Author Comment (AC1) · 15 May 2018

**Final author comments for "A continuum model of ice mélange and its role during retreat of the Antarctic Ice Sheet", by D. Pollard, R.M. DeConto and R.B. Alley.**

We thank the reviewers for their careful and helpful reviews. We agree with almost all of their comments and plan to act on nearly all of their specific suggestions for changes. Each comment, response and planned change are described below, with the reviewers' text in blue and ours in black.

**Referee: J.M. Amundson**

**Overview**

In this study the authors modify a set of equations used to model ice shelves in order to model the flow of ice mélange and to test its impacts on the behavior of the Antarctic Ice Sheet. Over the past decade or so, several observational papers have highlighted the relationship between ice mélange mobility and the seasonal advance and retreat cycle of marine-terminating glaciers. Few studies have attempted to model ice mélange, and only a couple have attempted to model ice mélange using continuum models. Development of continuum models of ice mélange is necessary for implementation into prognostic glacier/ice sheet simulations that commonly run for centuries to millenia. Thus, this paper is an important step toward implementing ice mélange into climate models.

My primary concerns revolve around the choice of the constitutive relationship for ice mélange. Essentially, the authors use Glen's Flow Law, which describes the viscous deformation of glacier ice, to model the motion of a granular material. The modifications that they make are based on knowledge of granular materials (little/no resistance to divergence) and sea ice (compression becomes difficult when the thickness becomes large). It's possible that these modifications are sufficient to yield a good approximation of ice mélange behavior, but I'm not sure. Unfortunately, there is very little data with which to test any constitutive relation for ice mélange, and so the approach adopted here seems about as good as any other, especially since the architecture is already in place for solving the flow equations. That said, the results from the model need to be interpreted with caution, which I think the authors have done.

We agree that the rheology is very uncertain, with only a few previous studies on ice mélange to draw on. As noted by the reviewer, the paper emphasizes this uncertainty and caution in interpreting these preliminary results, and the need for future work including connecting with results from discrete-particle studies (this is mostly in the Conclusions).

An additional uncertainty in the model is the lack of seasonality. In the model ice mélange generally exhibits extensional flow due to glaciostatic pressure gradients (similar to ice shelves). However, ice mélange flow in winter is often quasi-static, with all icebergs moving at the same speed as the glacier terminus. (In other words, pressure gradients are completely balanced by shear stresses along the fjord walls.) Ultimately, what this means is that the model equations used here probably underestimate ice mélange resistance in winter and overestimate it in summer. I would like to be convinced that this produces similar results to a model in which the ice mélange is really stiff in winter and weak in summer.

Mélange processes in Jakobshavn and other Greenland fjords clearly have strong seasonal dependence. In the paper we posit that our exploration of parameter space yields results that meaningfully capture net annual quantities and their dependence on forcing, without explicitly resolving the intra-annual variations. Seasonal variations are left to future work, as part of more detailed calibration for Jakobshavn (mentioned in the Conclusions). Note that adding seasonality would require seasonal variations in calving in the ice sheet model, which no large-scale long-term model has yet included to our knowledge. We do emphasize in the paper that the sub-seasonal episodic "pumping" of terminus advances and calving episodes on the

mélange is explicitly captured by the boundary condition described in Appendix B (Eq. B2), albeit for annual mean quantities.

On a related note, I'm not entirely sure if the boundary condition at the end of the mélange is correct, since it implies that gravitational spreading should occur even when the ice is very thin. Sea ice models are generally able to ignore this effect since sea ice typically moves very quickly; however, in models of landfast sea ice (e.g., Leppäranta, 2012) the constitutive relation is modified to ensure that gravitational spreading doesn't occur for thin ice by including a pressure term similar to the $P_p$ term that the authors use to inhibit compression of thick mélange. It may be that the boundary condition is fine since the authors aren't modeling winter mélange and are just trying to capture the net annual effect. I'm just not really sure how to think the boundary condition in this case.

As the reviewer says, this modification to sea-ice models, described in Leppäranta (2012) (who refers to Hibler, 2001), is designed to prevent deformation (stretching) right at the edge of the sea-ice pack where it is very thin, which is not observed. As the reviewer notes, it is not at all clear if this is relevant or desirable for mélange, which is not a solid material, and also does not thin to very small edge-values in the annual mean. However, we will add this point in the Conclusions as a possible concern for future work, and reference the above two papers.

**Comments**

- Please replace the reference to Amundson et al. (2016) with Burton et al. (2018). The former is not a peer-reviewed article, and the latter includes the same material (and more). A notable result of Burton et al., which is relevant here, is that ice mélange resistance is related exponentially to the ice mélange length-to-width ratio (*L/W*). (Note that we have also found similar results using a continuum approach and assuming a Coulomb-like rheology, though that work is currently in review.) Assuming parallel-sided fjords and ignoring any potential binding effects of sea ice, we found that ice mélange was only capable of inhibiting calving when *L/W* > 3. This is consistent with the results from this study, which indicate the ice mélange doesn't have much effect on the wide ice shelves found in Antarctica. It could be interesting to compare - what is the length-to-width ratio of fjords in Antarctica for which the model predicts that ice mélange will affect the rate of retreat?

  Thank you for this information on Burton et al. (2018). We will replace the Amundson et al. (2016) reference throughout. We will also add discussion on the *L/W* ratio in the Conclusions, as above - saying it is around 5 in our Jakobshavn simulations, and essentially 0 for resolved Antarctic embayments, so the contrast in our Greenland vs. Antarctic results is consistent with the Burton et al. relationship.

- I think the modifications to the constitutive relationship could be made more compact and transparent. I think you could write something like this:

$$\tau' = 2\,\eta\,\dot{\varepsilon}_{ij}\,(1 + (f\text{-}1)\,\delta_{ij}) - P_p\,\delta_{ij} \tag{1}$$

  As it currently stands, the reader has to get through some other details before reading about these modifications, which to me are really the most important parts of the paper. $P_p$ kind of appears out of nowhere, since it wouldn't naturally arise from vertical integration of the stress balance equations

unless it appears in the constitutive relation, and the description of $f$ is buried several paragraphs after it is first introduced.

By formulating the constitutive relation similarly to how I have here, you can immediately point out that (i) the equation is identical to the standard equations for glacier flow when $i \neq j$, (ii) there is low resistance to extension when $i = j$, and (iii) the $P_p$ term prevents $\dot{\varepsilon}_{xx}$ or $\dot{\varepsilon}_{yy}$ from becoming highly negative (compressive) when the mélange is thick.

We will move the description of $f$ to right after it is first used in Eq. 1. The existing description of $P_p$ already appears very close to and below Eq. (4) where it first appears. We disagree slightly with the reviewer on the inclusion of $P_p$ in his Eq. (1). Pressure terms appear in equations for the total stress $\tau$, but not for the deviatoric stress $\tau'$ (the difference $\tau'$-$\tau$ being the total pressure; e.g., Thoma et al., 2014, their Eqs. 5 and 7). Eq. (1) above and our Eq. (1) are for $\tau'$. Pressure gradients including $P_p$ appear naturally in the net force balance equations, as in our Eq. (4). We will slightly expand our description of $P_p$ after that equation, describing it as an "internal pressure" term augmenting hydrostatic pressure.

We could add the reviewer's Eq. (1) without the $P_p$ term, just before our existing Eq. (1). But then it would be exactly the same as our existing Eq. (1) except in $i,j$ notation, and we think that would just be extra material and not helpful for readers regarding the $f$ term. The main improvement for the readers will be describing $f$ right after Eq. (1), as the reviewer suggests.

**Detailed Comments**

P. 1, L. 10: Its not clear that ice mélange noticeably affects glacier velocities directly.

We will remove "velocities and", so the phrase will be: "which slows ice-front calving rates".

P. 2, L. 9: This statement makes it sound like ice mélange is often just icebergs connected by sea ice, and not a densely-packed granular material.

We will add "densely packed or", so the phrase will be: "Mélange consists of discrete ice pieces, densely packed or loosely cemented within sea ice" (in first bullet, section 2).

P. 3, L. 23: Also probably affected by currents, wind, and tides (which would have an asymmetric influence).

We will add a sentence "Currents, winds and tides may also move the mélange, but are assumed to be minor here." (in second bullet, section 2).

P. 4, L. 5: Should be Vankova and Holland.

Thank you - we will correct this reference.

P. 4, L. 25: I agree that large icebergs are never overriden, but there could be some small scale rotation occurring, resulting in storage of gravitational potential energy.

Right at the calving front, this may be significant for large overturning events observed at the termini of Jakobshavn and Helheim, but if it happens just there as mélange is being generated, we think it is effectively incorporated in the value of $H_n$ in Eq. B3, the thickness of newly created mélange. Farther downstream, if

the rotation occurs without contacting other bergs, there is no direct effect on the macro dynamics, except via waves. If it does contact other bergs, in principle that could be included in the internal pressure term $P_p$ by slightly modifying the form of its dependence on $h_m$ (Eq. 5) – but that would be beyond the scope of this initial basic model, left for future work.

**Referee: N. Golledge**

**Overview**

This paper presents a new and innovative approach to the simulation of ice melange as an integrated component of a continental-scale ice sheet model. The subject is topical, and such developments are necessary for ensuring that future-focused Greenland and Antarctic ice sheet simulations capture processes relevant to their likely evolution, particularly with regard to how each ice mass might contribute to changes in future sea level. The basic approach of the paper is to use and modify equations of the shallow shelf approximation to simulate the ice melange as a thin and poorly-aggregated 'ice shelf' that exerts some back force against the calving face of the true ice shelf. Taking a continuum approach rather than employing fracture mechanics enables the scheme to integrate more easily with the ice sheet / shelf model equations.

The paper is very well-written, methodical, and clear, it is well illustrated and the arguments are easy to follow. My comments are primarily suggestions that could better define the effects of the melange, which would implicitly address some of the 'uncertainty' in simulating this kind of poorly-observed process. Fundamentally my greatest concern is that there is a significant disconnect between the training scenario - a narrow Greenlandic fjord under present conditions - and the test scenario - the entire Antarctic ice sheet under a drastically warmer than present climate. Because of this, I feel that the Antarctic results, and the conclusions that depend on them, are not sufficiently convincing. By considering only two scenarios, the reader is left with no sense of any thresholds or dependencies in the system, which I feel are essential in terms of process understanding. I would like to see the following experiments added to help demonstrate the relative importance of the various model components in dictating how effective melange backpressure actually is:

1) Rather than just one 'Pliocene' scenario, why not use a range of step warmings in both atmosphere and ocean, with and without the melange feedbacks, to define the point at which melange becomes irrelevant? These should start at modern conditions and increment gradually, because at the moment the imposed 2deg C ocean warming is so high that it will almost certainly lead to rapid melt of the melange before any buttressing can exert an influence on GL location.

2) Following from that, there needs to be better separation of the model components, so a set of duplicate experiments are necessary in which, for a small or modest warming scenario, the effects of removing each of the important model components can be seen - i.e. - for a given ocean warming, show how GL retreat differs when a) ocean melt of the melange is turned off, b) shelf hydrofracture is turned off, c) tidewater cliff collapse is turned off. These kind of sensitivity experiments are alluded to in the conclusions, but not shown, which I think is a shame.

If these experiments all show that the inclusion of melange processes has no effect on GL retreat, then the conclusions of the paper will be a lot more robust, and the modelling community will be happy that we don't need to add such schemes to existing models.

We think these are very good points, encompassing one overall concern: that our results may only be applicable to very fast Antarctic retreats, accelerated by (i) step-function warming to Pliocene climate, and (ii) drastic mechanisms of hydrofracturing and cliff-failure (as needed to produce EAIS basin retreat). The issue is that mélange could conceivably play a more significant role in slower WAIS collapses ($O$(1000's) compared to $O$(100's) years), that have been found in other model studies without those mechanisms and with more gradual climate warming. The latter is of considerable interest to the modelling community, regarding whether mélange needs to be included in these models, or not.

We will address this by including one additional simulation. We think just one run is needed, not a suite of sensitivity runs, because all factors that can possibly allow the mélange to play more of a role can be included reasonably in one run. The modifications for this run are (1) a gradual linear ramp from modern to warm Pliocene climate over the first 300 years of the run (which is similar in timescale and Antarctic summer amplitude to future business-as-usual warming), (2) removal of hydrofracturing and cliff-failure mechanisms, and (3) zeroing of surface and basal melt on the mélange itself (which is unrealistic, but allows for possible overestimates of this melting). In previous experiments with our ice-sheet model and no mélange, this type of run produces slower WAIS collapse and no retreat in EAIS basins, as in the other model studies.

We have already performed this run, and the result is unequivocally that mélange still produces negligible back stress and slowing of ice-front retreat. So our main conclusion will still be that the much wider transverse scales of Antarctic basins (compared to Greenland fjords), and the unimpeded spreading of mélange into the Southern ocean, result in insignificant retardation of retreating ice fronts, even in scenarios with slower MISI-driven retreat. We will describe the additional simulation and say this in a new section 4.4 "Slower Antarctic retreat", and a new figure 10.

**A. Kerkweg**

Dear authors,

in my role as Executive editor of GMD, I would like to bring to your attention our Editorial version 1.1: http://www.geosci-model-dev.net/8/3487/2015/gmd-8-3487-2015.html This highlights some requirements of papers published in GMD, which is also available on the GMD website in the 'Manuscript Types' section: http://www.geoscientific-modeldevelopment. net/submission/manuscript_types.html In particular, please note that for your paper, the following requirement has not been met in the Discussions paper:

• "The main paper must give the model name and version number (or other unique Printer-friendly version Discussion paper identifier) in the title."

Please provide the name and the version number of the ice melange model in the title of your revised manuscript.

We will add a model name in the title: "A continuum model (PSUMEL1) of ice mélange and its role during retreat of the Antarctic Ice Sheet", and add the following sentence at the start of model description in section 3: "Our mélange model is labelled PSUMEL1 (Penn State University ice MELange model version 1)."

---

## Author Response (AR2)

Dear Didier Roche,

Thank you for taking over editorship of our manuscript in review for GMD, and your editorial comments. Thanks also to Jason Amundson for his second review. We have made the minor revisions as described below, and uploaded the new set of files to the GMD website.

For output and code access, I have archived the package of output and code to the Data Commons archive at Penn State, including a read-me file of the contents (README_PDA2018.txt). The specific url and doi that access this dataset are noted in the revised manuscript under "Code and Output Availability".

I don't know why the figures of the first-revised version did not come through - perhaps because there was a problem uploading them at the time and I had to do that twice. They are all the same as the original figures, except for:

- In Fig. 5a-b, which show multiple-ensemble Jakobshavn profiles, a thick brown curve is added in each panel for the case of very weak melange corresponding to Fig. 3d (as was noted in our first response-to-reviewers file). I had forgotten to include this case in the original Fig. 5.
- Fig. 10 was added as a whole new figure, in response to N. Golledge's review. It shows Antarctic results without the mechanisms of hydrofracturing and cliff-failure, and is discussed in section 4.4 (as was noted in our first response-to-reviewers file).

We have made the following minor edits in response to the two comments in J. Amundson's second review. His first comment (in italics) was:

*First, in some of the simulations it seems that the ice melange immediately adjacent to the terminus is thinner than the ice melange father down fjord (this is probably easiest to see in Figures 1 and 2). I'm not quite sure how to think of that, and it seems inconsistent with observations/intuition. A sentence or two addressing this issue would be helpful.*

In response, we added on pg. 8 lines 26-27 (for idealized channels): (The narrow strips very close to the ice face on the left of these figures are plotting artifacts; melange is actually thickest at the ice face and thins downstream).

We also edited and added, on pg. 10 lines 32-34 (for Jakobshavn, probably not what the reviewer referred to, but similar): As a consequence of the increased viscosity and basal drag, centerline velocities are only ~1 to 3 km yr-1. The constricted flow produces a bulging of thickness downstream from the grounding line that relies on along-flow stresses in the SSA equations (Eq. (4)) to maintain westward velocities.

J. Amundson's second comment was:

*Second, it would also be helpful if one of the panels in Figure 6 had some labels to indicate the basins that are referred to in the text. Not all readers will be familiar with Wilkes, Aurora, and Recovery Basins, etc.*

In response, we have added those labels and a few others in Fig. 6, first panel, asnoted in the figure caption.

These text edits are shown as track-changes in the marked-up manuscript version following below.

Sincerely,

David Pollard.

[revised manuscript text omitted]